
# A novel framework of deriving joint impoundment
# rules for large-scale reservoir system based on a
# classification-aggregation-decomposition approach
Shaokun He[1, 2, 3], Shenglian Guo[1, *], Chong-Yu Xu[1, 3], Kebing Chen[1], Zhen Liao[1], Lele Deng[1],
Huanhuan Ba[1], Dimitri Solomatine[2, 3]
[1]State Key Lab. of Water Resources & Hydropower Engineering Science, Wuhan Univ., 430072,
Wuhan, China;
[2]Hydro-informatics Group, IHE Delft Institute for Water Education, Delft, the Netherlands;
[3]Water Resources Sect., Delft Univ. Technol., Delft, the Netherlands;
[4]Dept. of Geosciences, University of Oslo, Oslo, Norway.
*Correspondence author*: *Shenglian Guo* (Email: slguo@whu.edu.cn).
**Abstract:** Joint and optimal impoundment operation of the large-scale reservoir system
has become more crucial for modern water management. Since the existing techniques
fail to optimize the large-scale multi-objective impoundment operation due to the
complex inflow stochasticity and high dimensionality, we develop a novel combination
of parameter simulation optimization and classification-aggregation-decomposition
approach here to overcome these obstacles. There are four main steps involved in our
proposed framework: (1) reservoirs classification based on geographical location and
flood prevention targets; (2) assumption of a hypothetical single reservoir in the same
pool; (3) the derivation of the initial impoundment policies by the non-dominated
sorting genetic algorithm-II (NSGA-II); (4) further improvement of the impoundment
policies via Parallel Progressive Optimization Algorithm (PPOA). The framework



potential is performed on China's mixed 30-reservoir system in the upper Yangtze River.
Results indicate that our method can provide a series of schemes to refer to different
flood event scenarios. The best scheme outperforms the conventional operating rule, as
it increases impoundment efficiency from 89.50% to 94.16% and hydropower
generation by 7.70 billion kWh (or increase 3.79%) while flood control risk is less than

0.06.

**Keywords:** Large-scale reservoir system; Joint impoundment rules; Multi-objective
operation; Classification-aggregation-decomposition; Yangtze River basin

## 1 Introduction

Rapid economic development and the growth of the human population are

responsible for more serious and wider water-related challenges, which lead to greater
stress on water resources management. One of the most effective measures to alleviate
water issues is to regulate natural streamflow via reservoirs (Lauri et al., 2012;Ng et al.,
2017). Cascade impoundment operation can properly achieve the goal since it stores
excess water during the wet season and depletes reservoir storage during the dry season
(Afshar et al., 2010;Labadie, 2004). In recent decades, impoundment operation has been
one hot academic topic. Considerable research efforts (Liu et al., 2011;Paredes and
Lund, 2006;Xu et al., 2017;Yeh, 1985) frequently point out that its key to the scientific
operation is deriving effective operating policies. However, most of the literature
merely focuses on the small-scale reservoir system, yet fails to address the complex
inflow stochasticity and high dimensionality of the multi-objective trans-basin (Yan et
al., 2012) and trans-province impoundment problems (Jurasz and Ciapala, 2017), even
if the latter large-scale impoundment operation is more necessary and suitable for



modern water resources management (Wang et al., 2014;Zhou et al., 2018).
As a matter of fact, the inherent scientific characteristics of inflow stochasticity
for large-scale impoundment operation has no difference with the small-scale one, there
are three theoretical breakthroughs to cope with it: (1) implicit stochastic optimization
(ISO) (Feng et al., 2017), (2) explicit stochastic optimization (ESO) (Goor et al., 2010),
and (3) parameter simulation optimization (PSO) (Zhang et al., 2019). ISO requires
'perfect inflow forecast' and ESO behaves in a more complex way to explicitly
incorporate all inflow probabilities. PSO is relatively preferred for large-scale operation
(Celeste and Billib, 2009;Tan et al., 2017), which predefines a rule curve shape and then
employs heuristic algorithms to identify the best parameter combination under all
possible inflow scenarios. Regarding another well-known 'high dimensionality' in the
PSO framework, the original simulation model is usually replaced by a surrogate model
for simplification. The surrogate should preserve and describe the main features of the
original model (Chu et al., 2015;Shaw et al., 2017;Zhang et al., 2017). The subtle
combination of the PSO framework and a surrogate model has indeed made some
achievements in addressing inflow stochasticity and dimensional curse of multi-
reservoir hydropower (Glotic and Zamuda, 2015;Valdes et al., 1992) and flood control
operations (Zhang et al., 2019), but is seldom utilized in large-scale impoundment
operation.
The major challenge lies in the reliability of the surrogate model. On the one hand,
it should highlight the reservoir storage state as the most vital indicator to track the
original system status, since the highest priority of impoundment operation is to ensure



certain storage for flood prevention during the operating horizon and to raise enough
end reservoir storage for water demand during the following dry period (Li et al.,
2018;Xu et al., 2017); Additionally, it also should reduce the number of decision
variables, making it possible to solve the curse of dimensionality. While it is noticed
that reservoirs can be classified into different pools according to the tributaries and
flood prevention targets (Zhang et al., 2014), a novel idea of 'classification-
aggregation-decomposition' is naturally introduced to structure the proper surrogate
model. The reservoirs in the same pool are firstly aggregated in water units to capture
reservoir storage information, then a decomposition method is used to decentralize
reservoir storage decisions into individual reservoirs in each pool. The salient feature
of this approach is to simplify the large-scale system into several equivalent hypothetic
reservoirs via aggregation, which caters to the replacement requirement of the
impoundment model.

Nevertheless, the current decomposition methods still have some degree of

drawbacks. Li et al. (2014a) and Zhang et al. (2019) allocated the virtual reservoir
output to individual reservoirs by using the empirical equations. Both made a quick
decision on decomposition forms but did not consider the maximum utilization of water
resources. Tan et al. (2017) adopted an improved genetic algorithm to seek for the
optimal decomposition scheme in the water-supply systems, but it is more time-
consuming since the calculation of the evaluation function goes to an exponential
increase with the number of involved reservoirs (Castelletti et al., 2012;Zhao et al.,
2012). These common techniques cannot balance computing efficiency and optimal



operating rules well. This limits their further application in practice. Actually, the recent
implementation of parallel computation has been proved able to reach a balance point
(Li et al., 2014b;He et al., 2019), although the parallelization technique attracts little
attention up to now. To this end, an emerging method-Parallel Progressive Optimization
Algorithm (PPOA) (Feng et al., 2018b) is introduced to assist our decomposition
strategy. It is a means of improving the quality of optimization while using a multi-core
configuration to enhance execution efficiency (Cheng et al., 2014).

To verify the feasibility of the proposed framework, we select a typical mixed 30-

reservoir system in China as the case study, where two objectives of reservoir
impoundment efficiency (*IE*) and flood control risk (*FCR*) are simultaneously
optimized, and then PPOA improves hydropower generation of individual reservoirs of
each pool without *IE* and *FCR* distortion. The remainder of the paper is structured as
follows: Section 2 addresses the 30-reservoir system and describes their conventional
operating rules for impoundment operation; Section 3 introduces the framework in
detail; Section 4 and Section 5 provide the experimental results and the application
prospects of this method; Section 6 ends with the conclusions.
**2 Case study**

The Yangtze River (in Fig. 1a) basin possesses abundant water potential in China.

It drains a catchment of 1.80 million km$^2$ with a total length of 6,300km. Its main
tributaries include the Jinsha River, Yalong River, Min River, Jialing River, and Wu
River. Owing to the subtropical monsoon climate, the Yangtze River basin often suffers
from the uneven temporal and spatial distribution of flood hazards induced by heavy



rainfalls. A series of large reservoirs have been built along its mainstream and tributaries
to allocate water resources in recent decades. Among them, a 30-reservoir system
including the core Three Gorges Reservoir (TGR) (in Fig. 1b) is one of the most critical
water conservancy projects in China. Most reservoirs in the system serve multi-
purposes (e.g., flood control, energy generation, tourism), except for Ge-Zhou-Ba (GZB)
which is a run-of-river hydropower station. Particularly, TGR serves as the largest water
project around the world, which is not only equipped with 22.50 GW installed
hydropower capacity, but also prevents downstream flood disaster. The 30-reservoir
system usually implements an impoundment operation to develop water resources. The
characteristic parameters of the 30 reservoirs are shown in Table 1.

<Please insert Fig. 1 here>

<Please insert Table 1 here>

For the impoundment operation of these reservoirs, Conventional Operating Rules

(CORs) are often individually predefined at the planning stage of reservoir
construction to provide guidelines. As Fig. 2 shows, COR suggests that the reservoir
water level should rise linearly throughout the whole impoundment horizon until it is
up to the top of the conservation pool. However, overlapping the impoundment time
of cascade reservoirs makes them compete with each other. It results in reduced water
supplies to face the following non-flood season. In fact, two preset strategies of (1)
advancing initial impoundment operation timing, but not earlier than Aug 1st; (2)
raising the reservoir water levels, can be activated to excavate the potential benefits of
joint impoundment operation on China's reservoir communities in the upper Yangtze



River basin (He et al., 2019). The aim of the joint impoundment operation of the 30-
reservoir system is to make efficient water resources utilization, on the condition that
it reserves enough storage capacity for flood control. The restored inflow series from
Aug 1st to Dec 31st spanning over 57 years (i.e. 1956-2012) are collected from the
Yangtze River (Changjiang) Water Resources Commission. The time step used is ten
(or eleven) days, a traditional Chinese measure of time, and therefore there are 15
operating periods for the five months per year.

<Please insert Fig. 2 here>

## 146   3 Methodology

Fig. 3 shows our research framework to derive the optimal impoundment rules for
the 30-reservoir system. The methodological modules are summarized below:
(1) All reservoirs are classified into different pools (in Table 1) according to their
geographic locations and flood prevention targets etc.;
(2) Reservoirs in the same pool are aggregated in water units to be a virtual
reservoir and the virtual reservoir storage is treated as the state variable during the
optimization process;
(3) PSO is employed to derive initial impoundment rules by considering the trade-
offs between *IE* and *FCR*;
(4) PPOA helps formulate the final impoundment rules by boosting hydropower
generation without *IE* and *FCR* distortion.

<Please insert Fig. 3 here>

### 159   3.1 Reservoirs classification



During an impoundment optimization process, the dimensionality of decision
variables increases linearly with the number of reservoirs; meanwhile, the
computational burden of the objective function trends an exponential increase (Galelli
and Castelletti, 2013). The vast reservoir community results in 'dimensionality disaster',
which makes it tricky to derive effective joint rules. An attractive alternative to solve
dimensionality restrictions is to classify these reservoirs and reduce their decision
variables. The basic idea is to divide the 30-reservoir system into different pools
according to their geographic distributions in the same tributary and flood prevention
targets etc. (Heever and Grossmann, 2000;Saad et al., 1994).

## 3.2 Aggregation and decomposition schemes

Conceptually, these reservoirs in the same pool can be assembled into a virtual one.
The hypothetical reservoir should retain the main characteristics yet ignore its hydraulic
connection (Duran et al., 1985). As shown below, the reservoirs are aggregated in water
units to simplify guidance on impoundment optimization (Tan et al., 2017):
$$V_i^*(t) = \sum_{n=1}^{M_i} V_{i,n}(t) \tag{1}$$

$$I_i^*(t) = \sum_{n=1}^{M_i} I_{i,n}(t) - Eva_i(t) \tag{2}$$

where $V_i^*(t)$ and $I_i^*(t)$ are hypothetical reservoir storage and inflow of the $i$th pool at
time $t$, respectively; $V_{i,n}(t)$ and $I_{i,n}(t)$ are the storage and inflow from external sources
of the $n$th reservoir of the $i$th pool at time $t$, respectively; $Eva_i(t)$ is the sum loss of the
$i$th pool at time $t$ (e.g., evaporation, seepage); $M_i$ is the total number of reservoirs in the
$i$th pool.



As reservoir storage is often treated as the state variable for reservoir operation
(Feng et al., 2018a;Chang and Chang, 2006), the virtual storage $V_i^*(t)$ is chosen here
as the state variable for the optimization process. It can be easily formulated by Eq. (1)
in any case of reservoir topology (shown in Fig. 4). Fig. 4(a) and Fig. 4(b) are
considered in our case study.

<Please insert Fig. 4 here>

The state variable $V_i^*(t)$ of each pool at all periods could be determined by the
aggregated impoundment rules. Another issue is how to allocate $V_i^*(t)$ to individual
reservoirs in the same pool. Some traditional decomposition strategies (e.g., fixed
proportions) have been experimented well (Turgeon, 1980;Zhang et al., 2019). In our
study, a similar decomposition way of the percentage of the allowable reservoir storage
is initially taken:
$$V_{i,n}(t) = SL_{i,n}(t) + (V_i^*(t) - \sum_{m=1}^{M_i} SL_{i,m}(t)) \times \frac{SS_{i,n}(t) - SL_{i,n}(t)}{\sum_{m=1}^{M_i} (SS_{i,m}(t) - SL_{i,m}(t))}$$
(3)

where $SL_{i,n}(t)$ is the lower boundary of the $n$th reservoir storage of the $i$th pool at time
$t$; $SS_{i,n}(t)$ is the $n$th reservoir storage of the $i$th pool at time $t$, which is relative to its
seasonal top of buffer pool.

**3.3 Parameter simulation optimization (PSO)**

The decomposition structure is embedded into the aggregation module, where the
latter combines with the multi-objective impoundment model to explore trade-offs
between the *IE* maximization and the *FCR* minimization (Liu et al., 2011;Zhou et al.,
2015). We established the PSO framework (Giuliani et al., 2016;Celeste and Billib,
2009) here to identify the initial impoundment strategies. As there are seven pools for





the 30-reservoir system and 15 decision variables for each virtual reservoir (one
decision variable corresponds to one operating period), there are a total of 105 (=7*15)
decision variables for these seven virtual reservoirs rather than 450 (=30*15) decision
variables in real.

### 3.3.1 Objective functions and constraints

*IE* is a critical indicator for impoundment operation to assess water resources
potential in the case of controllable *FCR* (Zhou et al., 2018):
(1) *IE* represents future water resources utilization for the following non-flood
period, it can be defined as follows:

$$max \ IE = \frac{1}{Y}\sum_{y=1}^{Y}\frac{\sum_{i=1}^{I}\sum_{n=1}^{M_i}(VE_{i,n}(y)-SD_{i,n})}{\sum_{i=1}^{I}\sum_{n=1}^{M_i}(SU_{i,n}-SD_{i,n})} \qquad (4a)$$

$$IE_{i,n} = \frac{1}{Y}\sum_{y=1}^{Y}\frac{VE_{i,n}(y)-SD_{i,n}}{SU_{i,n}-SD_{i,n}} \qquad (4b)$$

where $IE_{i,n}$ is the annual impoundment efficiency of the *n*th reservoir of the *i*th pool;
$VE_{i,n}(y)$ is the end storage of the *n*th reservoir of the *i*th pool in the *y*th year; $SU_{i,n}$, and
$SD_{i,n}$ is the *n*th reservoir storages of the *i*th pool, which corresponds to their top of
conservation pool and inactive pool, respectively. *I* and *Y* are the number of all pools
and years, respectively. *I* is 7 in our case.
(2) *FCR*, another critical objective to control the impoundment process, can be
evaluated as follows:

$$min \ FCR = min \ \{max \ \{FCR(t)\}\}, \quad (0 < t \le T \cdot Y) \qquad (5a)$$


$$FCR(t) = max\{\frac{\sum\limits_{i=1}^{I}\sum\limits_{n=1}^{M_i}(V_{i,n}(t)-SS_{i,n}(t))}{\sum\limits_{i=1}^{I}\sum\limits_{n=1}^{M_i}(SU_{i,n}-SS_{i,n}(t))},0\}$$ (5b)

$$FCR_{i,n}(t) = max\{\frac{V_{i,n}(t)-SS_{i,n}(t)}{\sum\limits_{i=1}^{I}\sum\limits_{n=1}^{M_i}(SU_{i,n}-SS_{i,n}(t))},0\}$$ (5c)

where $FCR_{i,n}(t)$ is the $FCR$ of the $n$th reservoir of the $i$th pool at time $t$; $T$ is the number
of operating periods in one year.
**3.3.2 Constraints**
A reservoir operation model generally contains the following constraints:
(1) Mass balance equation:
$$V_{i,n}(t+1) = V_{i,n}(t) + (I_{i,n}(t) + \sum_{j\in\Phi_{j,i,n}}R_j(t) - R_{i,n}(t))\cdot\Delta t$$ (6)

(2) Reservoir storage limits
$$SL_{i,n}(t) \le V_{i,n}(t) \le SS_{i,n}(t)$$ (7)

(3) Water discharge limits
$$RL_{i,n}(t) \le R_{i,n}(t) \le RU_{i,n}(t), \quad R_{i,n}(t) = Q_{i,n}(t) + QS_{i,n}(t)$$ (8)

(4) Hydropower generation limits
$$PL_{i,n}(t) \le N_{i,n}(t) \le PU_{i,n}(t)$$ (9)

(5) Boundary conditions
$$Z_{i,n}(t) = \begin{cases} Z_{i,n,begin}, & t=1,T+1,...,(Y-1)T+1 \\ Z_{i,n,end}, & t=T,2T,...,YT \end{cases}$$ (10)

where

| | |
|---|---|
| $V_{i,n}(t), V_{i,n}(t+1)$ | the $n$th reservoir storage of the $i$th pool at the beginning and end time $t$ |
| $\Phi_{j,i,n}$ | the upstream reservoir set which has a physical connection with the $n$th reservoir of the $i$th pool |
| $I_{i,n}(t)$ | the $n$th reservoir inflow of the $i$th pool from external sources at time $t$ |





| $R_{i,n}(t)$ | the $n$th reservoir release of the $i$th pool at time $t$ |
|---|---|
| $\Delta t$ | the time interval, day |
| $SL_{i,n}(t), SS_{i,n}(t)$ | the lower and upper storage boundaries of the $n$th reservoir of the $i$th pool at time $t$ |
| $RL_{i,n}(t), RU_{i,n}(t)$ | the lower and upper water release boundaries of the $n$th reservoir of the $i$th pool at time $t$ |
| $Q_{i,n}(t)$ | the generation discharge of the $n$th reservoir of the $i$th pool at time $t$ |
| $QS_{i,n}(t)$ | the spillway water of the $n$th reservoir of the $i$th pool at time $t$ |
| $PL_{i,n}(t), PU_{i,n}(t)$ | the lower and upper hydropower output boundaries of the $n$th reservoir of the $i$th pool at time $t$ |
| $Z_{i,n}(t)$ | the water level of the $n$th reservoir of the $i$th pool at time $t$ |
| $Z_{i,n,begin}, Z_{i,n,end}$ | the annual top of buffer pool and top of conservation pool of the $n$th reservoir of the $i$th pool |

**3.3.3 Optimization algorithm**
The NSGA-II algorithm (Deb et al., 2002) has made some successful
achievements in the PSO work of the reservoir field (Lei et al., 2018;Lotfan et al., 2016).
It realizes a fast convergence in Pareto frontiers with the crowding distance and the non-
dominated sorting rank (Deb et al., 2002). NSGA-II is implemented in this paper, even
if some other heuristic algorithms may also be able to handle it. The experimental
parameter is set as: the population size = 64, generation = 100, cross-over probability =
0.9 and mutation rate = 0.1.
**3.4 Coordination model**
The above procedures could realize a quick impoundment policy but fail to make
further water resource utilization. Finally, yet importantly, the problem is how to
excavate water potential.
**3.4.1 Objective function and constraints**



For large-scale impoundment operations in China, reservoir release usually
generates hydropower and then is transmitted to water consumers. As *IE* and *FCR*
occupy the highest priority, hydropower has to comply with the impoundment rules. A
good impoundment rule is to ensure that the impoundment quality of the reservoir is
achieved besides maximizing hydropower generation.
To this end, the corresponding coordination model is introduced below, which
maximizes the annual hydropower generation of the *i*th pool ($E_i$):

$$max \ E_i = \frac{1}{Y}\sum_{n=1}^{M_i}\sum_{t=1}^{T \cdot Y} N_{i,n}(t) \cdot \Delta t, \ N_{i,n}(t) = A_{i,n}Q_{i,n}(t)H_{i,n}(t) \tag{11}$$

where $A_{i,n}$ is the power coefficient of the *n*th reservoir of the *i*th pool; $H_{i,n}(t)$ is the
powerhead of the *n*th reservoir of the *i*th pool at time *t*; other symbols refer to Section

3.3.2.

Except for the constraints described in Section 3.3.2, the additional constraints in
Eq. 12 and Eq. 13(a-b) for each pool must be met during the whole period.

$$IE_i = \frac{1}{Y}\sum_{y=1}^{Y} \frac{\sum_{n=1}^{M_i}(VE_{i,n}(y) - SD_{i,n})}{\sum_{n=1}^{M_i}(SU_{i,n} - SD_{i,n})} \geq IE_i^* \tag{12}$$

$$FCR_i \leq FCR_i^* \tag{13a}$$

$$FCR_i = max\{\frac{\sum_{n=1}^{M_i}(V_{i,n}(t) - SS_{i,n}(t))}{\sum_{n=1}^{M_i}(SU_{i,n} - SS_{i,n}(t))}, 0\}, \ (0 < t \leq T \cdot Y) \tag{13b}$$

where initial $IE_i^*$ and $FCR_i^*$ of the *i*th pool are determined by Pareto Frontier in the
PSO framework.
**3.4.2 Parallel progressive optimization algorithm (PPOA)**
Recently, PPOA (Feng et al., 2018b) has emerged as a means of improving initial



solution quality. It can use abundant multi-core configuration to improve execution
efficiency while keeping the performance of the standard progressive optimization
algorithm (POA). The details of PPOA can be further referred to in other literature
(Feng et al., 2018b;Xie et al., 2015). Here is illustrated as an example of PPOA with 3-
reservoir and 3 levels per reservoir (in Fig. 5). Fig.5 shows that all the calculations of
the 27 (=$3^3$) state combinations are completely independent of each other for a single
sub-problem. In other words, the fitness value of any one state combination has no
influence on other state combinations, which reveals the good parallelism features.
PPOA adopts a successive approximation strategy to gradually improve solution quality,
which will make more sense with the dimensional expansion.

<Please insert Fig. 5 here>

## 4 Results

### 4.1 Pareto Frontiers of NSGA-II between *IE* and *FCR*

With the help of the NSGA-II algorithm, a wide array of Pareto Frontier is

explored within allowable impoundment storage capacity (in Table 2) and subsequently,
*COR* serves as the benchmark. The initial *IE* and *FCR* values of the extensively
distributed Pareto Frontier and the *COR* result for the whole research basin are
visualized in Fig. 6.

<Please insert Table 2 here>

<Please insert Fig. 6 here>

Fig. 6 shows that the improvement of one objective is bound to be followed by the

sacrifice of another objective. But these Pareto Frontiers still are able to adjustably
counterbalance *IE* and *FCR* simultaneously. More specifically, the optimal initial *FCR*





solution (Solution ①  in Fig. 6) can be considered while a wet inflow scenario during
the impoundment horizon is predicted in advance; on the contrary, the optimal initial
*IE* solution (Solution ③  in Fig. 6) could be more appropriate in a dry impoundment
scenario; at the end, the compromised initial solutions can be competent for different
impoundment scenarios with medium-scale inflow hydrograph.

We use the universal projection pursuit method (PP) (Friedman, 1987;He et al.,

2020) to evaluate the quality of all the Pareto Frontiers. PP can transform the 2-
dimensional values (*IE* and *FCR* values) to one-dimensional data with the help of the
projection vector and rank all solutions according to the one-dimensional value.
Because the possibility of large-scale flood events decreases over impoundment time,
these Pareto solutions with latter *FCR* appearance time are more preferred by decision-
makers. Finally, the Pareto Frontier (whose *IE* and *FCR* values are 94.16% and 0.06,
respectively) is selected as the initial optimal solution (Solution ②  in Fig. 6). Solution
①  and ③  are also included in the next sections, as Solution ①  always produces
much higher *IE* than COR when there is no flood control risk (*FCR* = 0); Solution ③
makes a better deal with joint impoundment operation in dry scenarios when the
influence of flood control can be ignored.
**4.2 Optimal results of the final impoundment policies**

We improved these three alternatives (i.e., Solution ①, ②,and ③) by the

PPOA and obtained the final three representative impoundment policies (I, II and III)
accordingly.
**4.2.1 Behavior performance between *IE* and *FCR***



The relationships between *IE* and *FCR* of the impoundment policies I, II, III as
well as COR for each of the 30 reservoirs are shown in Fig.7. Fig. 7 can easily
distinguish the numerical changes (i.e. the *IE* and *FCR* increment) of reservoirs. For
example, Reservoir D4 occupies the maximum *IE* value of 99.78% and Reservoir G1
occupies the maximum *FCR* value of 0.13. For Reservoir D4 with the maximum *IE*, it
illustrates that there is enough inflow in pool D to contribute to its small impoundment
storage, but upstream reservoirs with larger impoundment storage (Reservoirs D2 and
D3) in the same pool are difficult to fill up, even if their *FCR* value slightly increase.
Reservoir G1 has the maximum *FCR* value because Pool G fails to regulate massive
water from the upstream tributaries and large interval catchment area. Reservoir G1
also has an ideal *IE* result yet leaves inadequate storage to control flood risk, once it
activates the pre-set principle of raising the reservoir water level. In addition, it is
obvious that under the benchmark of COR, some of the other reservoirs can get better
*IE* improvement with a little or no *FCR* increase. For example, when the optimal
impoundment policy I is adopted, the *IE* increase of the four reservoirs in pool C varies
from 1.41% to 11.84%, while flood control standard remains unchanged (*FCR* is still
0). This means that: (1) these two pre-set strategies can have only positive effects on
impoundment operation when they are reasonably regulated; (2) the larger the required
impoundment storage of a reservoir, the better the potential impoundment prospect. The
sharp *IE* improvement of Reservoir G1 with the largest impoundment storage further
proves it. Reservoir G1 increases *IE* from 85.66% of COR to 94.64% of the
impoundment policy I (i.e. 1.99 billion m$^3$ increment of impoundment storage). By



contrast, the *IE* increment of reservoirs in pool A (except for Reservoir A6) and pool B
is relatively less, where these reservoirs with smaller impoundment storage are operated
by just lifting water level but remaining the initial impoundment timing unchanged.

<Please insert Fig. 7 here>

Fig. 7 also contains other significant information. Pools A, B, and F are not

sensitive to *FCR*. Their maximum *FCR* values are still 0 under our proposed
impoundment policies. It implies that these pools have enough flood control storage to
deal with relatively easier flood control tasks. With the implement of the impoundment
policy III, the other four investigative pools (C, D, E, and G) suffers from different
degrees of flood control risk, especially Reservoir G1 (*TGR*) in pool G. Aiming at *FCR*
varieties of Reservoir D2, D3 and D4 (in the same tributary) which are equipped with
synchronous impoundment operations (i.e. all the impoundment timings start at Sep.
20th and end on Oct. 31st), it shows that the maximum *FCR* value decreases from
upstream to downstream. The geographic elevations of these reservoirs have a
substantial influence on their *FCR* distribution with the impact of spare reservoir
storage. When runoff flows along cascade reservoirs, the upstream reservoir gives
priority to storage volumes increment to lift water levels, which causes a decrease of
downstream reservoir inflow. Consequently, the maximum *FCR* value of the
downstream reservoir can be obviously reduced. Nevertheless, with the staggered
impoundment time, these *FCR* values (e.g., Pool C) do not follow the principle of
sequential decrease of Pool D. The *FCR* varieties of these reservoirs equipped with
asynchronous impoundment operations in the same pool are more complicated to be





analyzed.

**4.2.2 Impact of optimal impoundment policies on hydropower generation**

In this study, hydropower generation is the only indicator to assess water resource
utilization of each pool. It is necessary to analyze the impact of the optimal
impoundment policies on hydropower generation with comparison to COR. For the
COR rule, the total simulated hydropower of the seven pools is 203.3 billion kWh. The
top three are Pool C, G, and A, whose proportions are 43.04%, 20.78% and 14.18% (in
Fig. 8), respectively. The sum of them is more than 75%. Hence, these three pools
should be more focused when the PPOA algorithm coordinates individual reservoirs.
<Please insert Fig. 8 here>
We set the maximum number of iterations of the PPOA algorithm for Pool A, C,
and G larger than other pools. Fig. 9 gives the hydropower generation results of seven
pools with three different impoundment rules. It indicates that the optimal
impoundment rules I, II and III can increase hydropower generation of the COR rule
by 3.11%, 3.79%, and 3.89%, respectively. The top three growth rates are Pool G, C,
and A. Especially, we can realize the huge potential hydropower of Pool G as its growth
rate ranging from 5.26% to 7.41%. The reason owes to that Pool G possesses abundant
water resources, where Reservoir G1 (TGR) is the largest hydropower plant in the word
and Reservoir G2 (GZB) is one run-of-river hydropower plant. They can generate
hydropower by converting kinetic energy into electricity more efficiently. Pool C can
also increase hydropower by 3.27%~4.07%. It increases so much hydropower outputs
because its four reservoirs located along the mainstream of the Yangtze River are





equipped with large installed hydropower capacity (47.81 GW in total, see Table 1),
thereinto, Reservoir C2 (BHT) is the second-largest hydropower station in China.
However, the growth rates of hydropower of the other five pools are not obvious, even
if there are six reservoirs in Pool A and seven reservoirs in Pool G. They stay low
efficiency from potential energy to electricity, since they store limited water resources
for future use.

<Please insert Fig. 9 here>

Moreover, Fig. 10 visualizes the hydropower increment of each pool in different

streamflow scenarios (assumed that one year represents one scenario). It shows more
hydropower details. It can be inferred that the optimal impoundment policy II is more
suitable for all scenarios in comparison with the impoundment I or III. Policy II and III
tap more hydropower potential of Pool C, G than I, but the flood control risk of II is
smaller than III. The hydropower increments of Pool A for policy II level off, which
illustrates that its impoundment rule is suitable for all the (wet, normal and dry)
scenarios. The hydropower increments of Pool B, D, E, and F incur a negative loss in
several dry scenarios, despite their annual average output present an increase. The slight
adverse changes in these dry years are due to when the upstream reservoirs in these
pools need more water to fill up their impoundment storage, the negative effect of the
reduced generation discharge overweight the positive effect of the raising water head
in the same time.

In addition, the hydropower growths of Pool C and G vary greatly with scenarios.

It further reveals that due to the complex hydraulic connections among the pools, the



single impoundment rule of Pool C and G cannot deal with all scenarios.

<Please insert Fig. 10 here>

**4.3 Other evaluation indicators**

Advanced joint impoundment operation will not only enhance water supply and

hydropower generation but also make other benefits including economy, $CO_2$ emission
reduction and so on, since it involves many factors directly or indirectly related to
impoundment efficiency (Zhou et al., 2018). Here we present the boxplot of the outflow
of all the seven pools (A~G) during the impoundment horizon to reveal its positive
influence on downstream.

Fig. 11 intuitively shows that the optimal impoundment policy II can improve

downstream streamflow requirements by altering its outflow distribution. For example,
it keeps the minimum downstream streamflow in Pool G no lower than 8000 $m^3$/s in
order to meet ecological needs, but the COR rule fails to satisfy this requirement in
some dry scenarios. Moreover, it ensures downstream streamflow no higher than 39,900
$m^3$/s for downstream flood control in most years and 54,000 $m^3$/s for its own flood
control safety in a few wet scenarios. It still behaves well than the COR rule to alleviate
pressure on downstream flood control, where its maximum downstream streamflow is
lower than that of the COR rule.

<Please insert Fig. 11 here>

**5 Discussion**

Due to the classification-aggregation-decomposition approach we put forward in

this study, the novel framework of deriving joint impoundment rules can effectively





overcome the tricky 'curse of dimensionality'. In order to explore the computational
efficiency of this method, we work in a Matlab environment equipped with a Windows
system (Intel[R] Core[TM] i5-4590 CPU @ 3.30 GHz and 8.00 GB of RAM) and list the
results of calculation time for different numbers of pools (shown in Fig. 12). Fig. 12
indicates that computational time increases with the number of pools, but does not
increase exponentially with the number of reservoirs. It owes to the fact that in our
method, the number of decision variables depends on the number of pools rather than
the number of reservoirs. Its outstanding performance will become more prominent as
the scale of the research reservoir is expanded. More specifically, time increases from
499 seconds to 8448 seconds when the number of reservoirs is from 6 to 30, which
provides new possibilities for optimizing the so vast reservoir community in 10-days
timescale. But actually, our proposed method can also optimize the daily impoundment
operation of the mixed 30-reservoir system by taking about 39700 seconds (almost 11
hours), when we make several experiments assuming that daily runoff value of each
reservoir can be discretized to be the same as its 10-day (11-day) runoff value.
<Please insert Fig. 12 here>
In addition, the novel method also inherits the inherent advantages of the multi-
objective optimization algorithm (NSGA-II here). Aiming at multi-objective reservoir
management, it not only produces the most optimal solutions as we referred to in
Section 4 but also generates a series of optimal strategies that can compromise well on
each objective. In our case, it gives rise to the operational alternatives available to
decision-makers in terms of the *IE* results.





Last but not least, we have to emphasize the benefits due to the introduction of the
PPOA mechanism. Compared to the final optimization results of traditional
decomposition strategies (i.e. Eq. (3)) (Zhang et al., 2017;Zhang et al., 2019), this
method can use PPOA to boost hydropower generation of the mixed 30-reservoir
system without *IE* and *FCR* distortion. The results between impoundment policies (I, II
and II) and the traditional optimal solutions (①, ②, and ③) are compared. The final
optimal *IE* and *FCR* results of impoundment policy I, II, III are 93.47% and 0, 94.16%
and 0.06, 94.22% and 0.14, respectively. It can be seen that the *FCR* value can further
be reduced (e.g., *FCR* of III is 0.14, less than that of Solution ③, 0.18). However, all
three policies increase hydropower generation to varying degrees: 209.65 billion of I vs
206.34 billion of ①, 211.02 billion of II vs 208.77 billion of ②, and 211.22 billion of
III vs 208.98 billion of ③. It also has a positive reference on future work of large-scale
flood control and hydropower operations.
**6 Conclusions**
This study attempts to structure an adept framework of deriving joint
impoundment rules, in which the classification-aggregation-decomposition and
parameter-simulation-optimization approach are coupled to deal with the complex
inflow stochasticity and high dimensionality in the large-scale cascade reservoirs, and
then PPOA algorithm further improves the performance of operating rules. With a case
study of the mixed 30-reservoir system in the upper Yangtze River, some vital
conclusions are summarized below:
(1) A large number of reservoirs can be classified into several pools and the



reservoirs in the same pool can be assembled to be an equivalent hypothetical reservoir.
It proves another feasible pathway to overcoming the classical dimensionality issue in
such a giant impoundment system via reducing decision variables.
(2) The multi-objective evolutionary algorithm coupled with the classification-
aggregation-decomposition approach has powerful capabilities for the complicated
cascade impoundment operation. With the help of the NSGA-II algorithm, the widely
distributed Pareto Frontiers enable water resources managers to favorably decide the
appropriate initial operating policies for a perfect compromise among the conflicting
objectives.
(3) The PPOA method can help further increase hydropower generation of each
pool without *IE* and *FCR* distortion. In comparison to the COR rule, our selected
optimal impoundment rule can increase reservoir impoundment efficiency from 89.50%
to 94.16% and hydropower generation by 7.70 billion kWh (or increase 3.79%) while
the flood control risk is less than 0.06.

**Data availability**
The inflow data and reservoir characteristics parameters of the 30-reservoir system
can be accessed by writing to the authors and filling a non-disclosure agreement under
certain conditions.

**Author contributions**
SLG and SKH conceived the original idea, and they designed the methodology.





SKH and KBC collected the data. SKH developed the model code and performed the
simulations, with some contributions from ZL, LLD, and HHB. SKH wrote the paper,
SLG, CYX, and DS revised the paper.

**Competing interests**


The authors declare that they have no conflict of interest.

**Acknowledgments**


We would like to thank the Yangtze River Commission for providing research data
for the reservoir community.

**Financial support**


Our sincere gratitude goes to the fundings from the National Key Research and
Development Plan (NO. 2016YFC0402206) of China, the National Natural Science
Foundation of China (NO. 51879192, NO. 51879194) and China Scholarship Council
(CSC).



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

Derivation of optimal joint operating rules for multi-purpose multi-reservoir water-
supply system, J Hydrol, 551, 253-264, 10.1016/j.jhydrol.2017.06.009, 2017.
Turgeon, A.: Optimal operation of multi-reservoir power-systems with stochastic
inflows, Water Resour Res, 16, 275-283, 10.1029/WR016i002p00275, 1980.
Valdes, J. B., Montbrundifilippo, J., Strzepek, K. M., and Restrepo, P. J.: Aggregation-
disaggregation approach to multireservoir operation, J Water Res Pl-Asce, 118, 423-
444, 10.1061/(Asce)0733-9496(1992)118:4(423), 1992.
Wang, X. M., Zhou, J. Z., Ouyang, S., and Li, C. L.: Research on joint impoundment
dispatching model for cascade reservoir, Water Resour Manag, 28, 5527-5542,
10.1007/s11269-014-0820-y, 2014.
Xie, M. F., Zhou, J. Z., Li, C. L., and Zhu, S.: Long-term generation scheduling of
Xiluodu and Xiangjiaba cascade hydro plants considering monthly streamflow



forecasting    error,    Energ    Convers    Manage,    105,    368-376,
10.1016/j.enconman.2015.08.009, 2015.
Xu, B., Boyce, S. E., Zhang, Y., Liu, Q., Guo, L., and Zhong, P. A.: Stochastic
programming with a joint chance constraint model for reservoir refill operation
considering flood risk, J Water Res Plan Man, 143, 10.1061/(Asce)Wr.1943-

5452.0000715, 2017.

Yan, D. H., Wang, H., Li, H. H., Wang, G., Qin, T. L., Wang, D. Y., and Wang, L. H.:
Quantitative analysis on the environmental impact of large-scale water transfer project
on water resource area in a changing environment, Hydrol Earth Syst Sc, 16, 2685-
2702, 10.5194/hess-16-2685-2012, 2012.
Yeh, W. W. G.: Reservoir management and operations models - a State-of-the-Art
review, Water Resour Res, 21, 1797-1818, 10.1029/WR021i012p01797, 1985.
Zhang, J. W., Wang, X., Liu, P., Lei, X. H., Li, Z. J., Gong, W., Duan, Q. Y., and Wang,
H.: Assessing the weighted multi-objective adaptive surrogate model optimization to
derive large-scale reservoir operating rules with sensitivity analysis, J Hydrol, 544, 613-
627, 10.1016/j.jhydrol.2016.12.008, 2017.
Zhang, J. W., Li, Z. J., Wang, X., Lei, X. H., Liu, P., Feng, M. Y., Khu, S. T., and Wang,
H.: A novel method for deriving reservoir operating rules based on flood classification-
aggregation-decomposition,    Journal    of    Hydrology,    568,    722-734,
10.1016/j.jhydrol.2018.10.032, 2019.
Zhang, R., Zhou, J. Z., Zhang, H. F., Liao, X., and Wang, X. M.: Optimal operation of
large-scale cascaded hydropower systems in the upper reaches of the Yangtze river,





China, J Water Res Plan Man, 140, 480-495, 10.1061/(Asce)Wr.1943-5452.0000337,

2014.

Zhao, T. T. G., Cai, X. M., Lei, X. H., and Wang, H.: Improved dynamic programming
for reservoir operation optimization with a concave objective function, J Water Res Pl-
Asce, 138, 590-596, 10.1061/(Asce)Wr.1943-5452.0000205, 2012.
Zhou, Y. L., Guo, S. L., Xu, C. Y., Liu, P., and Qin, H.: Deriving joint optimal refill
rules for cascade reservoirs with multi-objective evaluation, J Hydrol, 524, 166-181,
10.1016/j.jhydrol.2015.02.034, 2015.
Zhou, Y. L., Guo, S. L., Chang, F. J., and Xu, C. Y.: Boosting hydropower output of
mega cascade reservoirs using an evolutionary algorithm with successive
approximation, Appl Energ, 228, 1726-1739, 10.1016/j.apenergy.2018.07.078, 2018.





**List of Table**

**Table 1** Impounding times and characteristic parameters of the 30-reservoir system in
seven pools (A-G) in the upper Yangtze River
**Table 2** The sum storage capacity of $SS_{i,n}(t)$ of all reservoirs in the same pool at different
periods (billion m$^3$)






**Table 1**

Impounding times and characteristic parameters of the 30-reservoir system in seven pools (A–G) in the upper Yangtze River

| Pool | Reservoir | Initial time of impoundment | | End time of impoundment (COR, PSO) | Annual top of buffer pool (m) | Top of conservation pool[b] (m) | Total storage capacity (billion m³) | Storage capacity for flood control (billion m³) | Installed hydropower capacity (GW) |
| | | COR[a] | PSO[b] | | | | | | |
|---|---|---|---|---|---|---|---|---|---|
| Pool A[c] (middle Jinsha River) | (A1) LY | Aug. 1st | Aug. 1st | Sep. 30th | 1605 | 1618 | 0.81 | 0.17 | 2.40 |
| | (A2) AH | Aug. 1st | Aug. 1st | Sep. 30th | 1493.3 | 1504 | 0.89 | 0.22 | 2.00 |
| | (A3) JAQ | Aug. 1st | Aug. 1st | Sep. 30th | 1410 | 1418 | 0.91 | 0.16 | 2.40 |
| | (A4) LKK | Aug. 1st | Aug. 1st | Sep. 30th | 1289 | 1298 | 0.56 | 0.13 | 1.80 |
| | (A5) LDL | Aug. 1st | Aug. 1st | Sep. 30th | 1212 | 1223 | 1.72 | 0.56 | 2.16 |
| | (A6) GYY | Oct. 1st | Sep. 20th | Oct. 31st | 1128.8 | 1134 | 2.25 | 0.25 | 3.00 |
| Pool B (Yalong River) | (B1) LHK | Aug. 1st | Aug. 1st | Sep. 30th | 2845 | 2865 | 10.15 | 2.00 | 3.00 |
| | (B2) JP | Aug. 1st | Aug. 1st | Sep. 30th | 1859 | 1880 | 7.99 | 1.60 | 3.60 |
| | (B3) ET | Aug. 1st | Aug. 1st | Sep. 30th | 1190 | 1200 | 5.80 | 0.90 | 3.30 |
| Pool C (Downstream Jinsha River) | (C1) WDD | Aug. 10th | Aug. 1st | Sep. 10th | 952 | 975 | 3.94 | 2.44 | 10.20 |
| | (C2) BHT | Aug. 10th | Aug. 1st | Sep. 30th | 785 | 825 | 20.60 | 7.50 | 16.00 |
| | (C3) XLD | Sep. 1st | Aug. 20th | Sep. 30th | 560 | 600 | 12.67 | 4.65 | 13.86 |
| | (C4) XJB | Sep. 10th | Aug. 20th | Sep. 30th | 370 | 380 | 5.16 | 0.90 | 7.75 |
| Pool D (Min River) | (D1) ZPP | Oct. 1st | Sep. 20th | Oct. 31st | 850 | 877 | 1.11 | 0.17 | 0.76 |
| | (D2) XEX | Oct. 1st | Sep. 20th | Oct. 31st | 3105 | 3120 | 2.80 | 0.87 | 0.54 |
| | (D3) SJK | Oct. 1st | Sep. 20th | Oct. 31st | 2480 | 2500 | 2.90 | 0.66 | 2.00 |
| | (D4) PBG | Oct. 1st | Sep. 20th | Oct. 31st | 841 | 850 | 5.33 | 0.73 | 3.60 |





| | | | | | | | | | |
|---|---|---|---|---|---|---|---|---|---|
| Pool E (Jialing River) | (E1) BK | Oct. 1st | Sep.20th | Oct. 31st | 695 | 704 | 0.22 | 0.10 | 0.30 |
| | (E2) BZS | Oct. 1st | Sep.20th | Oct. 31st | 583 | 588 | 2.55 | 0.28 | 0.70 |
| | (E3) TZK | Sep. 1st | Aug. 20th | Sep. 30th | 447 | 458 | 4.07 | 1.44 | 1.10 |
| | (E4) CJ | Sep. 1st | Aug. 20th | Sep. 30th | 200 | 203 | 2.22 | 0.20 | 0.50 |
| Pool F (Wu River) | (F1) HJD | Sep. 1st | Aug. 20th | Sep. 30th | 1138 | 1140 | 4.95 | 0.15 | 0.60 |
| | (F2) DF | Sep. 1st | Aug. 20th | Sep. 30th | 968 | 970 | 1.02 | 0.04 | 0.57 |
| | (F3) WJD | Sep. 1st | Aug. 20th | Sep. 30th | 756 | 760 | 2.30 | 0.18 | 1.25 |
| | (F4) GPT | Sep. 1st | Aug. 20th | Sep. 30th | 628.1 | 630 | 6.45 | 0.20 | 3.00 |
| | (F5) SL | Sep. 1st | Aug. 20th | Sep. 30th | 435 | 440 | 1.59 | 0.18 | 1.05 |
| | (F6) ST | Sep. 1st | Aug. 20th | Sep. 30th | 357 | 365 | 0.92 | 0.21 | 1.12 |
| | (F7) PS | Sep. 1st | Aug. 20th | Sep. 30th | 287 | 293 | 1.47 | 0.23 | 1.75 |
| Pool G (TGR-GZB) | (G1) TGR | Sep. 10th | Aug. 20th | Oct. 31st | 145 | 175 | 45.07 | 22.15 | 22.50 |
| | (G2) GZB | - | - | - | - | 66 | 1.58 | - | 2.72 |

Note:

(a) COR: Conventional Operating Rule; (b) PSO: Parameterization Simulation Optimization.









**Table 2**

The sum storage capacity $SS_{i,n}(t)$ of all reservoirs in the same pool at different periods
(billion m$^3$)

| Pool | Aug.10$^{th}$ | Aug.20$^{th}$ | Aug.31$^{st}$ | Sep.10$^{th}$ | Sep.20$^{th}$ | Sep.30$^{th}$ | Oct.10$^{th}$ | Oct.20$^{th}$ | Oct.31$^{st}$ |
|------|------|------|------|------|------|------|------|------|------|
| A | 5.40 | 5.73 | 6.16 | 6.29 | 6.41 | 6.51 | 6.51 | 6.51 | 6.51 |
| B | 20.05 | 21.74 | 22.65 | 23.68 | 23.68 | 23.68 | 23.68 | 23.68 | 23.68 |
| C | 29.21 | 33.50 | 35.95 | 38.85 | 41.13 | 41.40 | 41.40 | 41.40 | 41.40 |
| D | 9.29 | 9.29 | 9.61 | 9.94 | 10.38 | 11.16 | 11.44 | 11.66 | 11.66 |
| E | 4.82 | 4.97 | 5.52 | 6.34 | 6.40 | 6.42 | 6.53 | 6.56 | 6.56 |
| F | 15.37 | 15.71 | 16.04 | 16.26 | 16.26 | 16.26 | 16.26 | 16.26 | 16.26 |
| G | 18.12 | 19.75 | 21.51 | 24.76 | 30.04 | 32.60 | 39.31 | 39.31 | 39.31 |

**Note:** the meaning of $SS_{i,n}(t)$ is referred in Eq. (3).



**List of Figures**




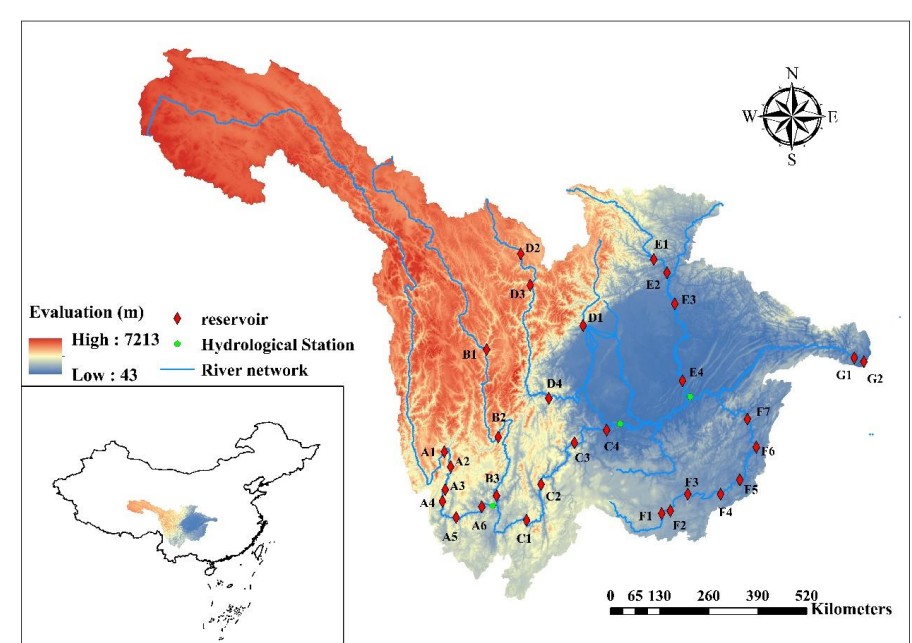

Fig. 1a Geographic distribution of the 30-reservoir system


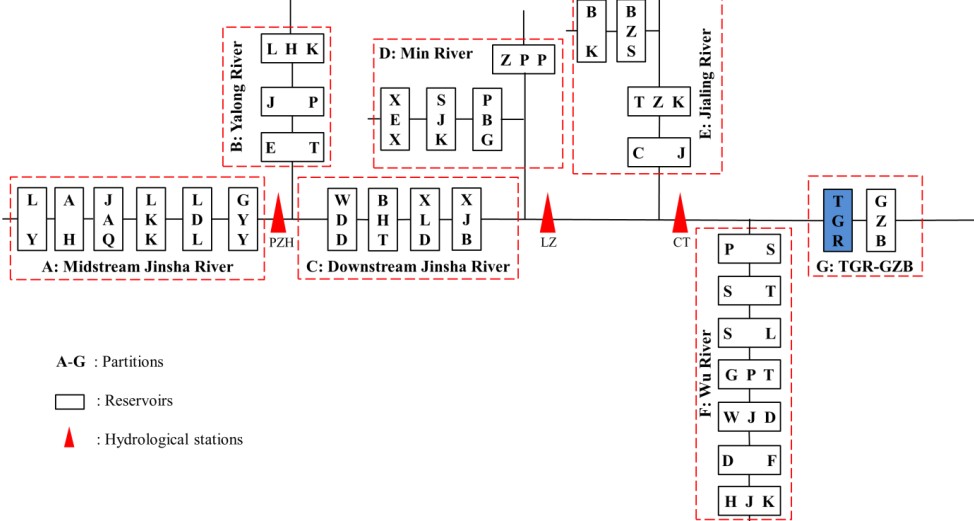

Fig. 1b Schematic diagram of the 30-reservoir system

**Fig. 1** Location information of the mixed 30 reservoirs in the upper Yangtze River



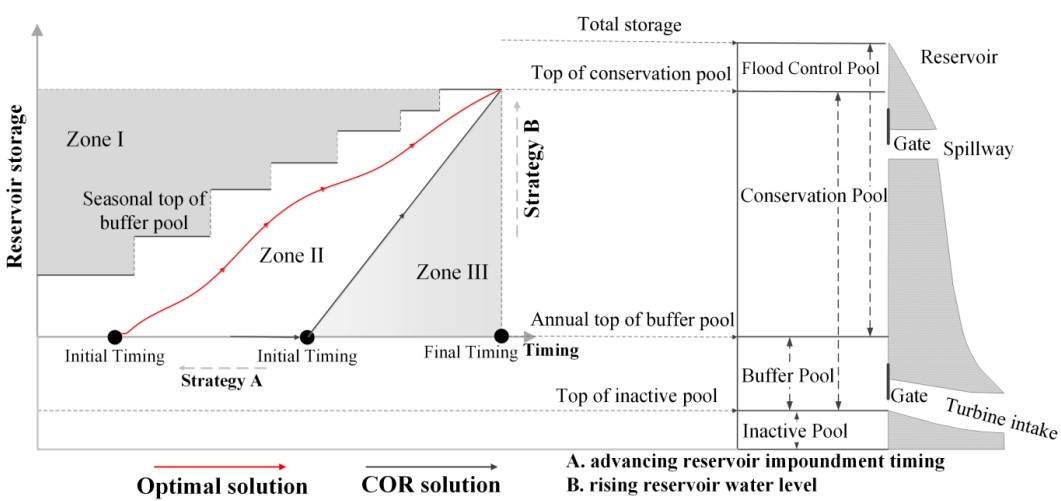

**Fig. 2** The schematic diagram of two activate strategies for advanced impoundment




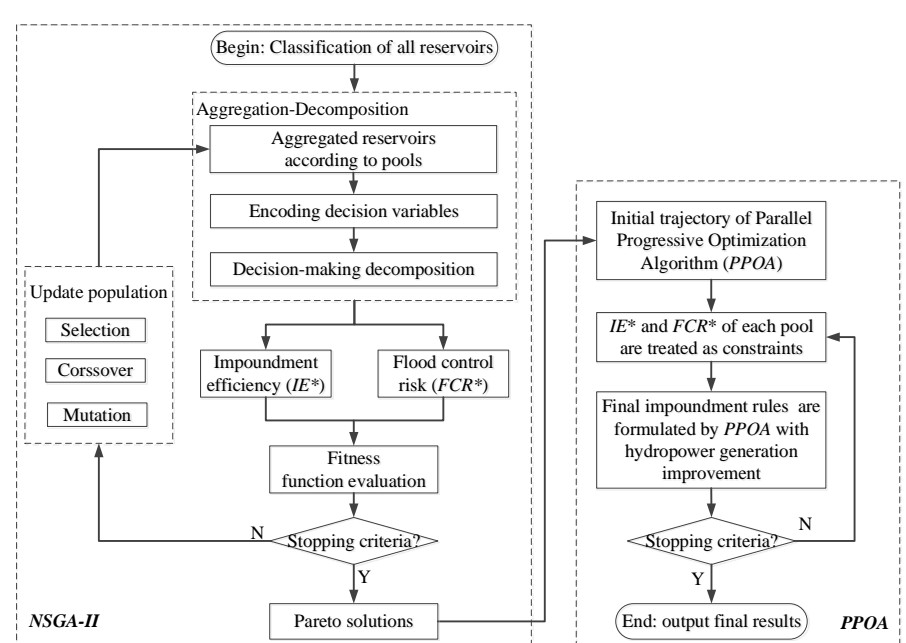

**Fig. 3** Flowchart of deriving joint impoundment rules for the large-scale reservoir
system




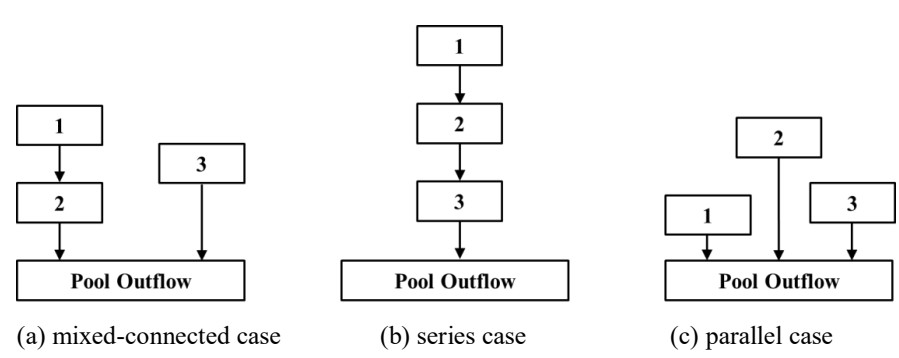

(a) mixed-connected case     (b) series case     (c) parallel case

**Fig. 4** Three general topological cases of reservoir location






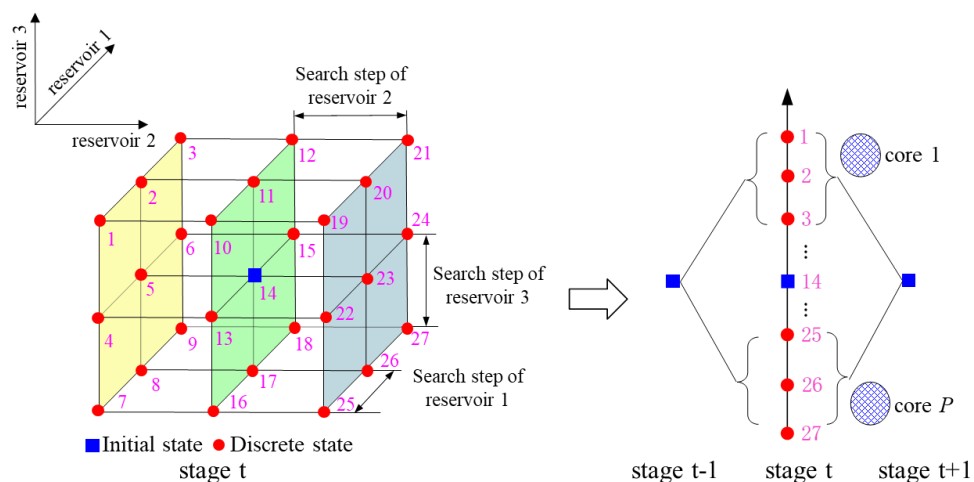

**Fig. 5** Sketch map of the PPOA algorithm for a sub-problem with 3-reservoir and 3
levels per reservoir







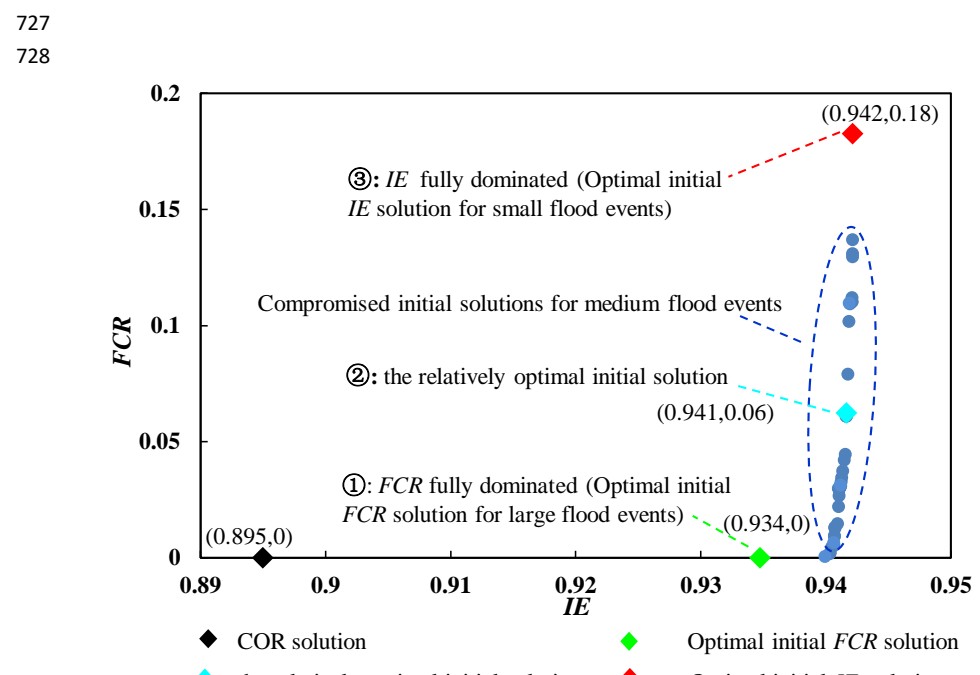


**Fig. 6** Comparison of the *IE* and *FCR* results of different initial optimal solutions





73

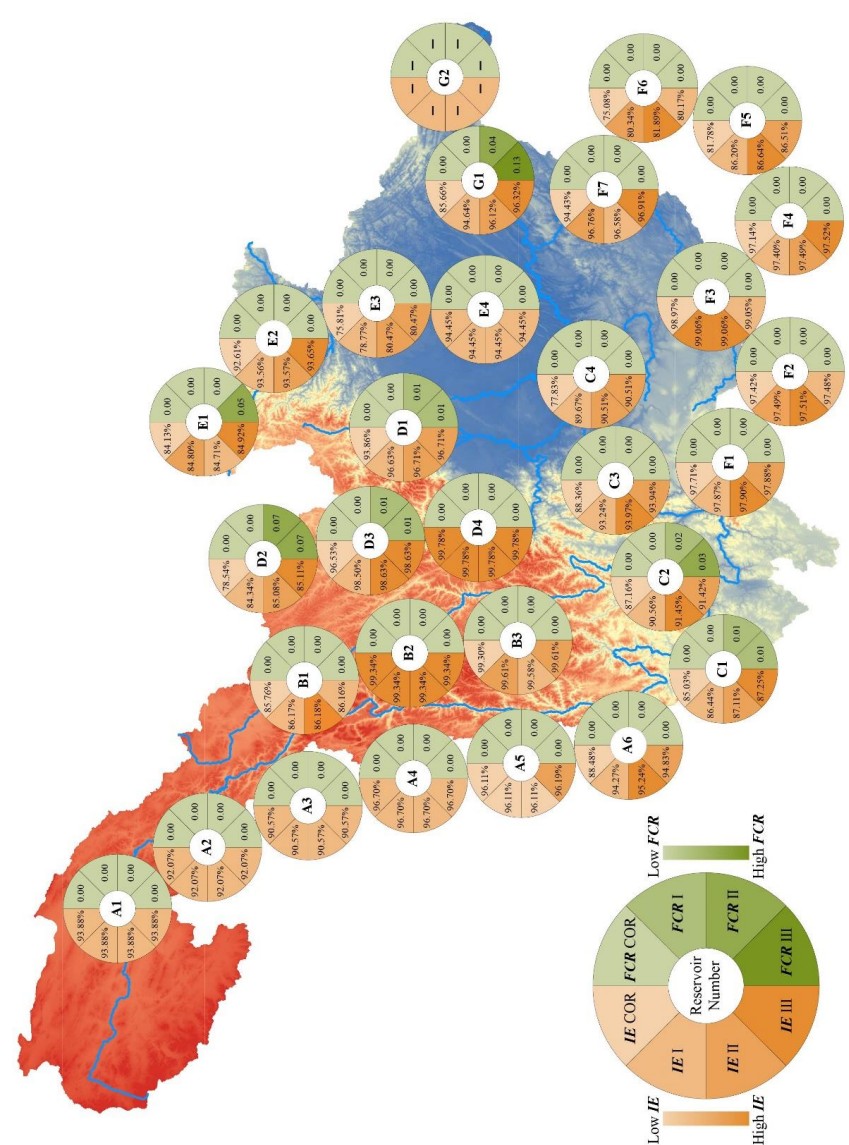

Fig. 7 Spatial varieties of *IE* and *FCR* of each reservoir relative to the different optimal impoundment rules




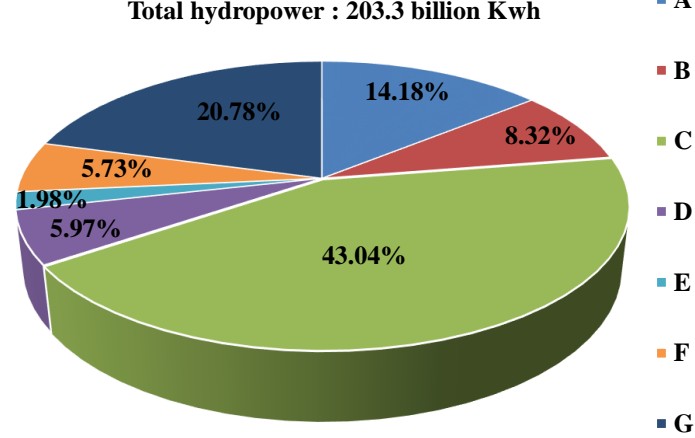


**Fig. 8** Hydropower proportion of each pool based on the COR rule






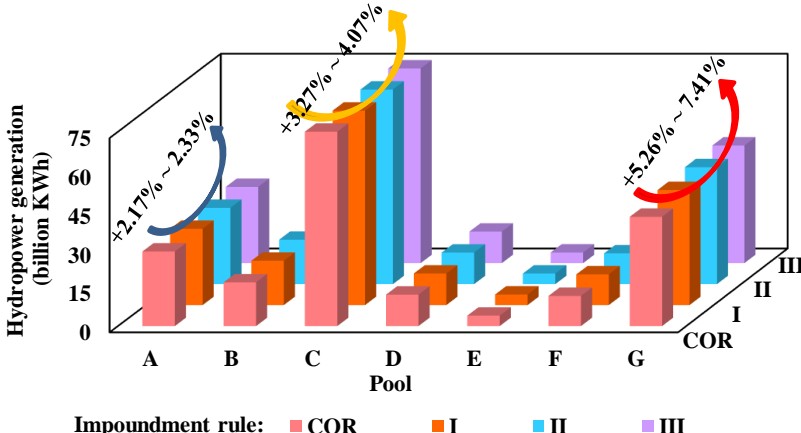


**Fig. 9** The hydropower results of the 30 reservoirs in seven pools for different impoundment rules




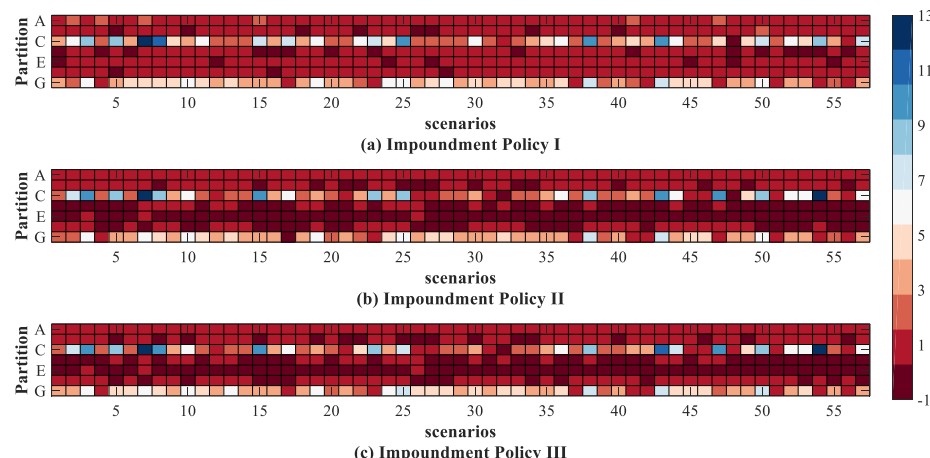

**Fig. 10** Hydropower increment of seven pools (A-G) for three optimal impoundment
policies compared to the COR rule in different streamflow scenarios (Unit: billion kWh)






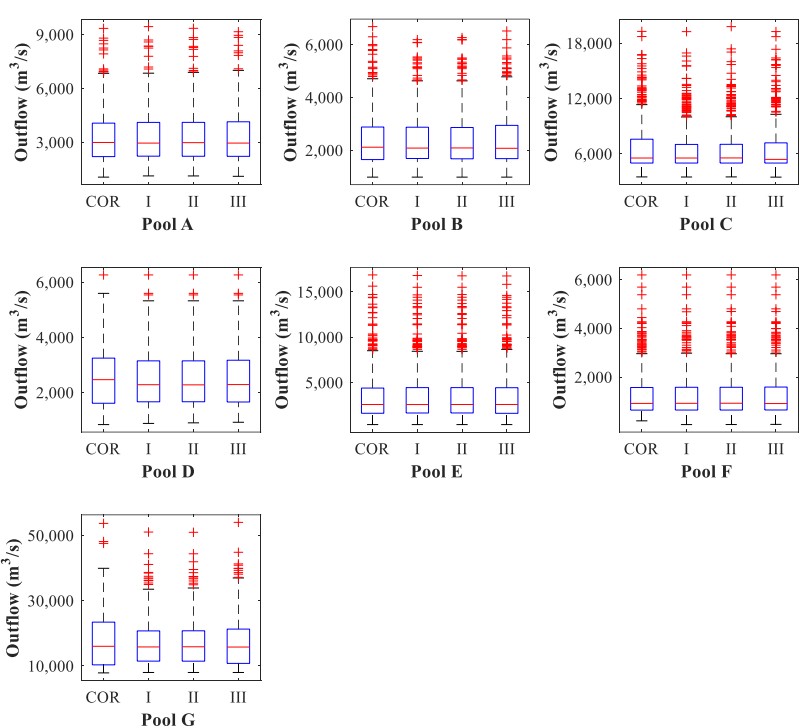


**Fig. 11** Outflow distribution of all Pools (A~F) during the impoundment period







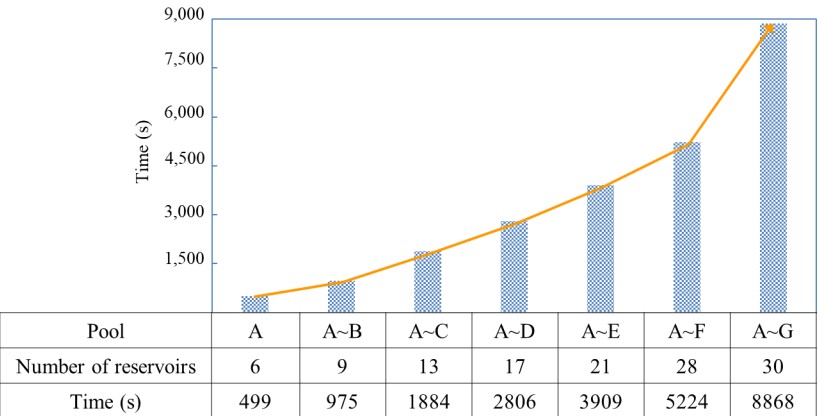

| Pool | A | A~B | A~C | A~D | A~E | A~F | A~G |
|---|---|---|---|---|---|---|---|
| Number of reservoirs | 6 | 9 | 13 | 17 | 21 | 28 | 30 |
| Time (s) | 499 | 975 | 1884 | 2806 | 3909 | 5224 | 8868 |


**Fig. 12** Computational efficiency for different numbers of pools