# Peer review of "A novel framework of deriving joint impoundment"

_Hydrology and Earth System Sciences, 2019_

## Referee Comment (RC1) · Anonymous Referee #1 · 14 Apr 2020

General comments

This manuscript describes a numerical optimization approach for the operation of large-scale water reservoir systems. Specifically, the authors attempt to tackle the so-called "curse of dimensionality", that is the fact that the computational requirements grow exponentially with the number of state (e.g., storage) and control (e.g., release decision) variables. Their approach relies on a classification-aggregation-decomposition scheme: first, reservoirs are classified and grouped based on location and operating targets; then, the problem of designing operating rules is solved for a smaller number

of hypothetic reservoirs representing the groups identified in the previous step. Finally, the decisions designed for these equivalent reservoirs are applied to the original system. The scheme is tested on a 30-reservoir network in the Yangtze River Basin.

I think that the topic of water resources management and reservoir operations is potentially relevant to the audience of HESS, but I do have several major concerns regarding this specific study.

Novelty. In my opinion, this work does not provide a novel, substantial contribution to the field of water reservoir operations. The idea of hierarchical multilevel decomposition has been around for a long time–see the work of Turgeon (1981), Saad and Turgeon (1988), or Archibald et al. (1997). Here, the authors just implement the same concept through the use of different optimization algorithms, such as NSGA-II and PPOA (whose description is rather unclear; see below).

Implementation and experimental setup. I believe the execution of this research presents some important flaws:

- The description of the inflow data is a concern here. First, the model requires inflow data for all reservoirs, but there appear to be only two gauging stations (see Figure 1a). How did the authors calculate the inflow to all reservoirs? (Unfortunately, the information provided at Line 140-144 is unclear.) Second, I would guess that not all reservoirs were built before 1956. So, how did you calculate the inflow to these reservoirs?

- The use of the current operating rules as a benchmark is likely to bias results and conclusions–a well-known concept in the water system analysis literature. That's because operators may not necessarily follow the objectives captured by the optimization problem. If the authors want to demonstrate that the proposed scheme advances the state-of-the-art, then they ought to compare it against existing optimization techniques.

- The optimization of the operating rules is carried out over the period 1956-2012; no data are used to validate these rules. This is likely to invalidate (or, at least, affect)

the conclusions: the genetic algorithm used to solve the problem is likely to "learn" decisions that work well only under these hydrological conditions. This is another well-known fact in the field of reservoir operations. In addition, this specific use of the inflow data does not seem to address the problem of "complex inflow stochasticity" mentioned in both Abstract and Introduction.

- There are no data showing that the experimental setup of NSGA-II is reliable (Line 245).

- There is no discussion of results and limitations.

Presentation. The overall quality of the presentation is very poor, largely below the standards of this journal. There are a few problems here:

- Many sentences contain grammatical mistakes or unclear and ambiguous statements, often preventing the reader from understanding entire paragraphs or sub-sections (see, for example, Section 3.4.2);

- The Introduction fails to explain some fundamental concepts, such as the "complex inflow stochasticity", which is mentioned throughout the manuscript;

- The manuscript, and in particular the Introduction, is not fully accessible to the audience of HESS, as it relies on a large amount of jargon that can only be grasped by the (narrower) reservoir operations community (see my detailed comments);

- There are some ambiguous / unclear / wrong statements that are likely to accidentally deceive the readers. See, for example, Line 45-50: while it is true that several studies on reservoir operations focus on small systems, it must also be acknowledged that there are dozens of works targeting the curse of dimensionality using either hierarchical multilevel decomposition or functional approximation (see Castelletti et al., 2010, and references therein).

Overall, I praise the attempt to tackling a difficult reservoir operation problem, but I believe that the limited novelty, unreliable results, and poor presentation provide ample

ground for recommending a rejection.

Specific comments

- Line 16: "the large-scale reservoir system". Which system?

- Line 17-19: In what terms do the existing techniques fail?

- Line 19: "high dimensionality". High dimensionality of . . .?

- Line 26-27: Check grammar.

- Line 28-29: What do you mean here?

- Line 30: "89.50% to 94.16%" of . . .?

- Line 32: "0.06". What is this variable? How is it measured?

- Line 43-44: do not add references in the middle of a sentence. It's better to use them to prove / consolidate a point you are trying to make.

- Line 51-55: The Introduction should be accessible to a broad audience in hydrology. The use of these concepts does not make it possible.

- Line 55-57: What does this mean?

- Line 60-62: Again, what does this mean? Make your thoughts available to the audience–do not rely on unknown jargon.

- Line 67-68: "seldom" implies that there have been some applications. Can you add these important references?

- Line 75-79. What does this mean?

- Line 87: Which "empirical equations"? What does this mean?

- Line 87-89: What does this mean?

- Line 97-98: If you want readers to follow you, you must explain the underlying concept

of PPOA.

- Line 105: "without IE and FCR distortion". What does this mean?

- Line 132-233: What does this mean?

- Line 144: "for the five months per year"?

- Line 163: "The vast reservoir community results in 'dimensionality disaster'. What does this mean?

- Line 160-168. This is just a repetition of previously-stated concepts.

- Line 248-250: what does this mean?

- Line 262: Is this supposed to be a new section?

- Line 300: You should explain what the "universal projection pursuit method" does.

References

Archibald, T., K. McKinnon, and L. Thomas (1997), An aggregate stochastic dynamic programming model of multireservoir systems, Water Resour. Res., 33(2), 333–340.

Castelletti, A., S. Galelli, M. Restelli, and R. Soncini‐Sessa (2010), Tree‐based reinforcement learning for optimal water reservoir operation, Water Resour. Res., 46, W09507

Saad, M., and A. Turgeon (1988), Application of principal component analysis to long‐term reservoir management, Water Resour. Res., 24(7), 907–912.

Turgeon, A. (1981), A decomposition method for the long‐term scheduling of reservoirs in series, Water Resour. Res., 17(6), 1565–1570.

---

## Author Comment (AC1) · 28 Apr 2020

Anonymous Referee #1 General comments This manuscript describes a numerical optimization approach for the operation of large-scale water reservoir systems. Specifically, the authors attempt to tackle the so-called 'curse of dimensionality', that is the fact that the computational requirements grow exponentially with the number of state (e.g., storage)

[Figure]

and control (e.g., release decision) variables. Their approach relies on a classification-aggregation-decomposition scheme: first, reservoirs are classified and grouped based on location and operating targets; then, the problem of designing operating rules is solved for a smaller number of hypothetic reservoirs representing the groups identified in the previous step. Finally, the decisions designed for these equivalent reservoirs are applied to the original system. The scheme is tested on a 30-reservoir network in the Yangtze River Basin. I think that the topic of water resources management and reservoir operations is potentially relevant to the audience of HESS, but I do have several major concerns regarding this specific study.

Reply: Thanks for your positive general evaluation of our work and constructive suggestions. We have carefully studied them and will revise the paper following these advices and suggestions. In the following, we provide point-by-point responses to your concerns and how we should address them in the revision.

Novelty. In my opinion, this work does not provide a novel, substantial contribution to the field of water reservoir operations. The idea of hierarchical multilevel decomposition has been around for a long time–see the work of Turgeon (1981), Saad and Turgeon (1988), or Archibald et al. (1997). Here, the authors just implement the same concept through the use of different optimization algorithms, such as NSGA-II and PPOA (whose description is rather unclear; see below).

Reply: Many thanks for your comments and sorry that we failed to explain clear enough of the novelty of our study in the previous version. We will try to explain it here and will also revise the paper accordingly. The concept of 'aggregation-disaggregation' method was proposed to solve the 'curse of dimensionality' in 1980s, and has been applied in different fields, e.g., water supply in Tan et al. (2017), flood control in Zhang et al. (2017, 2019). However, there are no effective methods to drive the giant reservoir impoundment system (Zhou et al., 2018). From the point of the impoundment operation of a 30-reservoir group in the upper Yangtze River basin, we firstly tried to build a model to aim at impoundment efficiency but also to consider other conflicting objectives (i.e.,

flood control risk and hydropower).

We identify the objectives of impoundment efficiency and flood control risk of the 30-reservoir group as the highest priority. This is a typical multi-objective optimization problem that can be solved using the 'classification-aggregation -decomposition' approach. As hydropower generation is regarded as the second priority, we employed the PPOA method to quickly optimize hydropower without the degradation of the results of impoundment efficiency and flood control risk. The double-layer simulation-based optimization model is the core novelty of our research work. In order to highlight the novelty of our work, we will revise the manuscript, especially the Title, Abstract, Introduction and Figure 3 (the flowchart of our work).

References:

[1] Tan, Q. F., Wang, X., Wang, H., Wang, C., Lei, X. H., Xiong, Y. S., and Zhang, W.: Derivation of optimal joint operating rules for multi-purpose multi-reservoir water-supply system, J Hydrol, 551, 253-264, 10.1016/j.jhydrol.2017.06.009, 2017.

[2] Zhang, J. W., Wang, X., Liu, P., Lei, X. H., Li, Z. J., Gong, W., Duan, Q. Y., and Wang, H.: Assessing the weighted multi-objective adaptive surrogate model optimization to derive large-scale reservoir operating rules with sensitivity analysis, J Hydrol, 544, 613-627, 10.1016/j.jhydrol.2016.12.008, 2017.

[3] Zhang, J. W., Li, Z. J., Wang, X., Lei, X. H., Liu, P., Feng, M. Y., Khu, S. T., and Wang, H.: A novel method for deriving reservoir operating rules based on flood classification-aggregation-decomposition, Journal of Hydrology, 568, 722-734, 10.1016/j.jhydrol.2018.10.032, 2019.

[4] Zhou, Y. L., Guo, S. L., Chang, F. J., and Xu, C. Y.: Boosting hydropower output of mega cascade reservoirs using an evolutionary algorithm with successive approximation, Appl Energ, 228, 1726-1739, 10.1016/j.apenergy.2018.07.078, 2018.

Implementation and experimental setup. I believe the execution of this research

presents some important flaws:

- The description of the inflow data is a concern here. First, the model requires inflow data for all reservoirs, but there appear to be only two gauging stations (see Figure 1a). How did the authors calculate the inflow to all reservoirs? (Unfortunately, the information provided at Line 140-144 is unclear.) Second, I would guess that not all reservoirs were built before 1956. So, how did you calculate the inflow to these reservoirs?

Reply: Many thanks for your professional comments. And again we realized that something was not clearly presented here. We only marked two representative hydrological stations in Figure 1a. Actually, there are many hydrological stations in this study basin. In the revision, we will improve Figure 1a.

In addition, these reservoirs were built in different years, it is impossible to calculate inflow dataset of these reservoirs from 1956-2012. As referred in Lines 140-142 and Lines 501-503: Acknowledgment, the restored inflow runoff data provided by the Yangtze River (Changjiang) Water Resources Commission of The Ministry of Water Resources are used in our calculation. In the revision, we will provide more detailed information about description of the input data.

- The use of the current operating rules as a benchmark is likely to bias results and conclusions–a well-known concept in the water system analysis literature. That's because operators may not necessarily follow the objectives captured by the optimization problem. If the authors want to demonstrate that the proposed scheme advances the state-of-the-art, then they ought to compare it against existing optimization techniques.

Reply: Thanks for your comments. In the real case of impoundment operation, the current operating rule is designed to guide operators to control water release. It also has a benchmark value of impoundment efficiency, hydropower and flood control risk. It is necessary to explore whether impoundment efficiency and hydropower can be improved or not when flood control risk is low. The results of current operating rule are usually included and compared (Lei et al., 2018; Li et al., 2018; Zhang et al., 2019;

Zhou et al., 2018).

In addition, we agree with your comment that our proposed method should be compared against some classical optimization techniques. We compared the novel combination method of aggregation-decomposition and PPOA with the aggregation-decomposition method as referred in Lines 450-461. In fact, there are no existing impoundment optimization techniques that can optimize the 30-reservoir group (He et al., 2019; Zhou et al., 2018). That's why we propose the novel framework to conquer the 'curse of dimensionality' problem. We will make changes of the manuscript to make it clearer.

References:

[1] He, S. K., Guo, S. L., Chen, K. B., Deng, L. L., Liao, Z., Xiong, F., and Yin, J. B.: Optimal impoundment operation for cascade reservoirs coupling parallel dynamic programming with importance sampling and successive approximation, Adv Water Resour, 131, 10.1016/j.advwatres.2019.07.005, 2019.

[2] Lei, X., Zhang, J., Wang, H., Wang, M., Khu, S. T., Li, Z., & Tan, Q. (2018). Deriving mixed reservoir operating rules for flood control based on weighted non-dominated sorting genetic algorithm II. Journal of Hydrology, 564, 967-983.

[3] Li, H., Liu, P., Guo, S. L., Ming, B., Cheng, L., and Zhou, Y. L.: Hybrid two-stage stochastic methods using scenario-based forecasts for reservoir refill operations, J Water Res Plan Man, 144, 10.1061/(Asce)Wr.1943-5452.0001013, 2018.

[4] Zhang, J. W., Li, Z. J., Wang, X., Lei, X. H., Liu, P., Feng, M. Y., Khu, S. T., and Wang, H.: A novel method for deriving reservoir operating rules based on flood classification-aggregation-decomposition, Journal of Hydrology, 568, 722-734, 10.1016/j.jhydrol.2018.10.032, 2019.

[5] Zhou, Y. L., Guo, S. L., Chang, F. J., and Xu, C. Y.: Boosting hydropower output of mega cascade reservoirs using an evolutionary algorithm with successive approxima-

tion, Appl Energ, 228, 1726-1739, 10.1016/j.apenergy.2018.07.078, 2018.

- The optimization of the operating rules is carried out over the period 1956-2012; no data are used to validate these rules. This is likely to invalidate (or, at least, affect) the conclusions: the genetic algorithm used to solve the problem is likely to "learn" decisions that work well only under these hydrological conditions. This is another well-known fact in the field of reservoir operations. In addition, this specific use of the inflow data does not seem to address the problem of "complex inflow stochasticity" mentioned in both Abstract and Introduction.

Reply: Thanks for your professional comment, which allows us to clarify here and also in the revision that the procedure strictly follows the commonly used Parameterization-Simulation-Optimization (PSO) theory to derive operating rules. The PSO theory, the implicit stochastic optimization (ISO) and explicit stochastic optimization (ESO) methods are three main ways to derive optimal operating rules, which is referred in Lines 52-60. The PSO has been used by many researchers (Celeste and Billib, 2009; Giuliani et al., 2016; Ostadrahimi et al., 2012; Zhang et al., 2019), and is different with hydrological model evaluation which divides dataset into the calibration and validation periods. The optimization of the operating rules is always carried out over the whole historical period.

We will explain it clearer that, as a matter of fact, the PSO theory has considered inflow stochasticity in its internal procedure. In other words, the optimal operating rules derived by PSO is reliable to deal with possible future inflow scenarios (Koutsoyiannis and Athanasis, 2003; Liu et al., 2011). We are sorry that we did not make it clear enough which might cause misunderstanding of the concept of 'inflow stochasticity' in Abstract and Introduction, and we will revise this part carefully.

References:

[1] Celeste, A. B., and Billib, M.: Evaluation of stochastic reservoir operation optimization models, Adv Water Resour, 32, 1429-1443, 10.1016/j.advwatres.2009.06.008,

2009.

[2] Giuliani, M., Li, Y., Cominola, A., Denaro, S., Mason, E., and Castelletti, A.: A mat-lab toolbox for designing multi-objective optimal operations of water reservoir systems, Environ Modell Softw, 85, 293-298, 10.1016/j.envsoft.2016.08.015, 2016.

[3] Koutsoyiannis, D., and Athanasia E.: Evaluation of the parameterization‐simu-lation‐optimization approach for the control of reservoir systems, Water Resources Research 39.6, 2003.

[4] Liu, X. Y., Guo, S. L., Liu, P., Chen, L., and Li, X. A.: Deriving optimal re-fill rules for multi-purpose reservoir operation, Water Resour Manag, 25, 431-448, 10.1007/s11269-010-9707-8, 2011.

[5] Ostadrahimi, L., Mariño, M. A., & Afshar, A. Multi-reservoir operation rules: multi-swarm PSO-based optimization approach. Water resources management, 26(2), 407-427, 2012.

[6] Zhang, J. W., Li, Z. J., Wang, X., Lei, X. H., Liu, P., Feng, M. Y., Khu, S. T., and Wang, H.: A novel method for deriving reservoir operating rules based on flood classification-aggregation-decomposition, Journal of Hydrology, 568, 722-734, 10.1016/j.jhydrol.2018.10.032, 2019.

- There are no data showing that the experimental setup of NSGA-II is reliable (Line 245).

Reply: We will explain it clearer by referring the relevant literatures that the experi-mental setup of NSGA-II is reliable (Yang et al., 2016, 2017; Zhang et al., 2019; Zhou et al., 2018). We set the relatively large values of population size and generation to guarantee convergence, and will make more instruction about the parameter settings.

References:

[1] Yang, G., Guo, S. L., Li, L. P., Hong, X. J., and Wang, L.: Multi-objective operating

rules for Danjiangkou reservoir under climate change, Water Resour Manag, 30, 1183-1202, 10.1007/s11269-015-1220-7, 2016.

[2] Yang, G., Guo, S. L., Liu, P., Li, L. P., and Liu, Z. J.: Multiobjective cascade reservoir operation rules and uncertainty analysis based on PA-DDS algorithm, J Water Res Plan Man, 143, 10.1061/(Asce)Wr.1943-5452.0000773, 2017.

[3] Zhang, J. W., Li, Z. J., Wang, X., Lei, X. H., Liu, P., Feng, M. Y., Khu, S. T., and Wang, H.: A novel method for deriving reservoir operating rules based on flood classification-aggregation-decomposition, Journal of Hydrology, 568, 722-734, 10.1016/j.jhydrol.2018.10.032, 2019.

[4] Zhou, Y. L., Guo, S. L., Chang, F. J., and Xu, C. Y.: Boosting hydropower output of mega cascade reservoirs using an evolutionary algorithm with successive approxima-tion, Appl Energ, 228, 1726-1739, 10.1016/j.apenergy.2018.07.078, 2018.

- There is no discussion of results and limitations.

Reply: Thanks for your comments. In the old version we discussed the impoundment results of the 30-reservoir system in Section 4 – Results, and discussed our novel method applied in cascade impoundment operation in Section 5 – Discussion. It is true that we failed to discuss the limitations of our proposed method, which will be paid more attention in the revision.

Presentation. The overall quality of the presentation is very poor, largely below the standards of this journal. There are a few problems here: - Many sentences con-tain grammatical mistakes or unclear and ambiguous statements, often preventing the reader from understanding entire paragraphs or sub-sections (see, for example, Sec-tion 3.4.2);

Reply: Many thanks for your comments. We are sorry for the unclear statements. We will carefully check the revised version, and we will also seek for help from professional language editing service for proofreading and language polishing before submitting the

revised version.

- The Introduction fails to explain some fundamental concepts, such as the "complex inflow stochasticity", which is mentioned throughout the manuscript;

Reply: Many thanks for your comments. We will carefully revise and improve the writing of the Introduction section, which is a very important part of a scientific paper.

- The manuscript, and in particular the Introduction, is not fully accessible to the audience of HESS, as it relies on a large amount of jargon that can only be grasped by the (narrower) reservoir operations community (see my detailed comments);

Reply: Many thanks for your comments, after reading the paper and reviewer's comment, we fully agree that the Introduction section needs a careful improvement to serve the wide community of audiences of HESS.

- There are some ambiguous / unclear / wrong statements that are likely to accidentally deceive the readers. See, for example, Line 45-50: while it is true that several studies on reservoir operations focus on small systems, it must also be acknowledged that there are dozens of works targeting the curse of dimensionality using either hierarchical multilevel decomposition or functional approximation (see Castelletti et al., 2010, and references therein).

Reply: Many thanks for your comments. We are sorry that the description in this paragraph is not comprehensive and somehow biased.

We agree that there are dozens of works targeting the curse of dimensionality in reservoir operation (Castelletti et al., 2010; Zhang et al., 2014). Even in recent years, curse of dimensionality is still studied in lots of literatures (Tan et al., 2017; Zhang et al., 2017 and 2019. but they are involved in flood control, hydropower or others rather than impoundment operation, including Castelletti et al. (2010). As impoundment operation of cascade reservoirs is also crucial in water resources management but few is studied (Zhou et al., 2018), we develop this research topic in our paper. We will revise the

manuscript to make it comprehensive, informative and balanced, to better represent the state-of-the-art of the topic dealt in the study.

References:

[1] Castelletti, A., S. Galelli, M. Restelli, and R. Soncini-Sessa (2010), Tree-based reinforcement learning for optimal water reservoir operation, Water Resour. Res., 46, W09507

[2] Tan, Q. F., Wang, X., Wang, H., Wang, C., Lei, X. H., Xiong, Y. S., and Zhang, W.: Derivation of optimal joint operating rules for multi-purpose multi-reservoir water-supply system, J Hydrol, 551, 253-264, 10.1016/j.jhydrol.2017.06.009, 2017.

[3] Zhang, J. W., Wang, X., Liu, P., Lei, X. H., Li, Z. J., Gong, W., Duan, Q. Y., and Wang, H.: Assessing the weighted multi-objective adaptive surrogate model optimization to derive large-scale reservoir operating rules with sensitivity analysis, J Hydrol, 544, 613-627, 10.1016/j.jhydrol.2016.12.008, 2017.

[4] Zhang, J. W., Li, Z. J., Wang, X., Lei, X. H., Liu, P., Feng, M. Y., Khu, S. T., and Wang, H.: A novel method for deriving reservoir operating rules based on flood classification-aggregation-decomposition, Journal of Hydrology, 568, 722-734, 10.1016/j.jhydrol.2018.10.032, 2019.

[5] Zhang, R., Zhou, J. Z., Zhang, H. F., Liao, X., and Wang, X. M.: Optimal operation of large-scale cascaded hydropower systems in the upper reaches of the Yangtze river, China, J Water Res Plan Man, 140, 480-495, 10.1061/(Asce)Wr.1943-5452.0000337, 2014.

Overall, I praise the attempt to tackling a difficult reservoir operation problem, but I believe that the limited novelty, unreliable results, and poor presentation provide ample ground for recommending a rejection.

Reply: We appreciate your careful review, and we apologize for some unclear expression in terms of novelty, clearness and readability. We will do a rigorous revision

following the advices of all the reviewers and editors.

Specific comments

- Line 16: "the large-scale reservoir system". Which system?

Reply: We will change the expression of "the large-scale reservoir system" to "the large-scale cascade reservoir group" in the revision.

- Line 17-19: In what terms do the existing techniques fail? - Line 19: "high dimensionality". High dimensionality of : : :?

Reply: reply to above two comments on lines 17-19. What we meant was "the existing techniques fail to optimize the large-scale multi-objective impoundment operation in limited time due to complex inflow stochasticity and high dimensionality of release decisions, we develop a novel combination of parameter simulation optimization and classification-aggregation-decomposition approach here to overcome these obstacles". We will rewrite this sentence in the revision.

- Line 26-27: Check grammar.

Reply: we meant: A case study is performed with a mixed 30-reservoir group in the upper Yangtze River basin. We will carefully check grammar for the whole manuscript by ourselves as well as by professional language editing service.

- Line 28-29: What do you mean here?

Reply: We tried to express that our selected operating rule is superior compared to the conventional operating rule. We will revise this sentence for clarity.

- Line 30: "89.50% to 94.16%" of : : :?

Reply: Thanks for your comments. We will correct it.

- Line 32: "0.06". What is this variable? How is it measured?

Reply: Thanks for your comments. The definition of flood control risk is referred in

Equation (5a) and (5b) in Line 221-222. We will revise this sentence for clarity.

- Line 43-44: do not add references in the middle of a sentence. It's better to use them to prove / consolidate a point you are trying to make.

Reply: Thanks for your comments. We will correct it in the revision.

- Line 51-55: The Introduction should be accessible to a broad audience in hydrology. The use of these concepts does not make it possible.

Reply: Many thanks for your comment, as responded earlier we will largely revise and improve the presentation of Introduction section to make it to serve the wide community of audiences of HESS.

- Line 55-57: What does this mean?

Reply: Thanks for your comment. We are sorry for the unclearness. What we meant was the three optimization methods (i.e., implicit stochastic optimization, ISO; explicit stochastic optimization, ESO; parameter simulation optimization, PSO) were commonly used in reservoir research (Celeste and Billib, 2009; Giuliani et al., 2016; Li et al., 2018; Zhang et al., 2019), while ISO and ESO are more complex than PSO in large-scale reservoir optimization. This point is what we want to convey in Lines 55-60. We will rewrite the sentence in the revision to make it clearer.

References:

[1] Celeste, A. B., and Billib, M.: Evaluation of stochastic reservoir operation optimization models, Adv Water Resour, 32, 1429-1443, 10.1016/j.advwatres.2009.06.008, 2009.

[2] Giuliani, M., Li, Y., Cominola, A., Denaro, S., Mason, E., and Castelletti, A.: A matlab toolbox for designing multi-objective optimal operations of water reservoir systems, Environ Modell Softw, 85, 293-298, 10.1016/j.envsoft.2016.08.015, 2016.

[3] Li, H., Liu, P., Guo, S. L., Ming, B., Cheng, L., and Zhou, Y. L.: Hybrid two-stage

stochastic methods using scenario-based forecasts for reservoir refill operations, J Water Res Plan Man, 144, 10.1061/(Asce)Wr.1943-5452.0001013, 2018.

[4] Zhang, J. W., Li, Z. J., Wang, X., Lei, X. H., Liu, P., Feng, M. Y., Khu, S. T., and Wang, H.: A novel method for deriving reservoir operating rules based on flood classification-aggregation-decomposition, Journal of Hydrology, 568, 722-734, 10.1016/j.jhydrol.2018.10.032, 2019.

- Line 60-62: Again, what does this mean? Make your thoughts available to the audience–do not rely on unknown jargon.

Reply: Thanks for your comments. We will change the sentence as follows:

Regarding the well-known 'high dimensionality' in the PSO, the original simulation model can be simplified by a surrogate model that preserves the main features of the original model.

- Line 67-68: "seldom" implies that there have been some applications. Can you add these important references?

Reply: Thanks for your professional comments. We will add one related reference in the revision.

Zhou, Y. L., Guo, S. L., Chang, F. J., and Xu, C. Y.: Boosting hydropower output of mega cascade reservoirs using an evolutionary algorithm with successive approximation, Appl Energ, 228, 1726-1739, 10.1016/j.apenergy.2018.07.078, 2018.

- Line 75-79. What does this mean?

Reply: Thanks for your comment. We will revise the sentence in the revised manuscript as follows:

As reservoirs can be classified into different pools according to their geographic distributions and flood prevention targets (Zhang et al., 2014), 'classification-aggregation-decomposition' is employed in this work:

- Line 87: Which "empirical equations"? What does this mean? - Line 87-89: What does this mean?

Reply: Thanks for your comments. We will rewrite the related sentences as "Li et al. (2014a) and Zhang et al. (2019) allocated the virtual reservoir output to individual reservoirs by using an empirical method without considering the maximum utilization of water resources."

- Line 97-98: If you want readers to follow you, you must explain the underlying concept of PPOA.

Reply: Thanks for your suggestion. As the concept of PPOA is very important for understanding our work, we need pay more attention to explaining it well. So, we will introduce it in detail in the revision.

- Line 105: "without IE and FCR distortion". What does this mean?

Reply: The term means that there is no decease of IE result and no increase of FCR. We will make a supplementary explanation in the revised manuscript.

- Line 132-233: What does this mean?

Reply: We will correct the sentence in the revision as "However, cascade reservoirs are often difficult to fill due to overlapping impoundment periods with limited inflow resources".

- Line 144: "for the five months per year"?

Reply: Thanks for your comment. We will delete this expression and rewrite this sentence in the revision.

- Line 163: "The vast reservoir community results in 'dimensionality disaster'. What does this mean?

Reply: Thanks for your comment. We will revise the sentence in the revision as "The
huge computational burden makes it difficult to derive effective joint operation rules."

- Line 160-168. This is just a repetition of previously-stated concepts.

Reply: Thanks for your comments. It will be reformulated.

- Line 248-250: what does this mean?

Reply: We will revise the sentences as follows:

The above procedures could identify an effective impoundment policy but cannot further improve hydropower generation. The potential of increasing hydropower generation based on above analysis is exploited as follow.

- Line 262: Is this supposed to be a new section?

Reply: No, '3.3.2' was a continuation from the sentence in line 261. It was "... other symbols refer to Section 3.3.2".

- Line 300: You should explain what the "universal projection pursuit method" does.

Reply: Thanks for your advice. We will explain the "universal projection pursuit method" in detail in the revision.

References Archibald, T., K. McKinnon, and L. Thomas (1997), An aggregate stochastic dynamic programming model of multireservoir systems, Water Resour. Res., 33(2), 333–340.

Castelletti, A., S. Galelli, M. Restelli, and R. Soncini⅘A ĚĞ RSessa (2010), Tree-based reinforcement learning for optimal water reservoir operation, Water Resour. Res., 46, W09507

Saad, M., and A. Turgeon (1988), Application of principal component analysis to long⅘ARĚĞ term reservoir management, Water Resour. Res., 24(7), 907–912.

Turgeon, A. (1981), A decomposition method for the long⅘ARĚĞ term scheduling of reservoirs in series, Water Resour. Res., 17(6), 1565–1570.

Please also note the supplement to this comment:
https://www.hydrol-earth-syst-sci-discuss.net/hess-2019-586/hess-2019-586-AC1-
supplement.pdf

―――――――――――――――

---

## Referee Comment (RC2) · Anonymous Referee #2 · 30 Apr 2020

The presented research, in my opinion, well fits the scope of the HESS and could be an important contribution to the management of large scale multi-objective reservoirs. In their very interesting paper, the authors presented a novel methodology that integrates the 'classification-aggregation-decomposition' and the Parallel Progressive Optimization Algorithm method to conquer the 'curse of dimensionality' problem of complex 30-reservoir impoundment operation. The idea of deriving a systematic optimization framework for an entire basin by taking into account multi-purposes all 30-reservoir is very attractive and ambitious. However, the current form of the manuscript possesses

several shortcomings which should be revised to improve its readability and quality.

(1) It is very hard annoying to review a paper where the figures are put at the end of the manuscript. It would be easier to review the manuscript if figures are as close as possible to the place in the text (and not upside down) where they are being referenced. (2) The methodology does not well present, especially fails to emphasize the function of the PPOA method in the framework. The authors should add more detailed information about PPOA to make it clear and informative. Fig 3 could be revised to explicitly show the four main steps involved in the proposed framework. (3) The schematic diagram of the PPOA algorithm in Fig 5 should be better drawing. (4) The comparison results of three optimal operating rules with the Conventional Operating Rule, in my opinion, seem tedious (for example, Figure 7), and a long explanation of Section 4.3 is not necessary. (5) Fig. 10, "Hydropower increment of seven pools for three policies compared to the COR in different streamflow scenarios", should be re-design and/or changed colors and make it more attractive and informative. (6) Some jargon in the reservoir community is not accessible to all audiences of HESS, ex., 'Inflow stochasticity' in implicit stochastic optimization (ISO), explicit stochastic optimization (ESO), and parameter simulation optimization (PSO) in the Introduction. (7) The references could be updated. (8) Proofreading by a native English speaker should be conducted to improve both language and organization quality.

---

## Author Comment (AC2) · 5 Jun 2020

The presented research, in my opinion, well fits the scope of the HESS and could be an important contribution to the management of large scale multi-objective reservoirs. In their very interesting paper, the authors presented a novel methodology that integrates the 'classification-aggregation-decomposition' and the Parallel Progressive Optimization Algorithm method to conquer the 'curse of dimensionality' problem of complex 30-reservoir impoundment operation. The idea of deriving a systematic optimization framework for an entire basin by taking into account multi-purposes all 30-reservoir is very attractive and ambitious. However, the current form of the manuscript possesses several shortcomings which should be revised to improve its readability and quality.

Reply: Thanks for your careful review and positive evaluation of our work. We will revise the paper following your professional advices and suggestions. In the following, we provide point-by-point responses to your concerns and how we should address them in the revision.

(1) It is very hard annoying to review a paper where the figures are put at the end of the manuscript. It would be easier to review the manuscript if figures are as close as possible to the place in the text (and not upside down) where they are being referenced.

Reply: In the revised version, we will put the figures in the text where they are referred

to.

(2) The methodology does not well present, especially fails to emphasize the function of the PPOA method in the framework. The authors should add more detailed information about PPOA to make it clear and informative. Fig 3 could be revised to explicitly show the four main steps involved in the proposed framework.

Reply: Thanks for your professional comments, we will emphasize the use of *PPOA* in the revision, especially Fig. 3, the flowchart of our methodology. In the revised version, Fig. 3. will read like the following:

[Figure]

Fig. 3 The framework of the large-scale reservoir impoundment system for policy optimization.

(3) The schematic diagram of the PPOA algorithm in Fig 5 should be better drawing.

Reply: Thanks for your comments, we will re-design this figure in the revision as follows.

[Figure]

Fig. 5 Sketch map of the *PPOA* algorithm for solution improvement at time *t*.

(4) The comparison results of three optimal operating rules with the Conventional Operating Rule, in my opinion, seem tedious (for example, Figure 7), and a long explanation of Section 4.3 is not necessary.

Reply: Thanks for your comments, we will streamline the comparison of different rules, as well as the description of Section 4.3 in the revised paper.

(5) Fig. 10, "Hydropower increment of seven pools for three policies compared to the COR in different streamflow scenarios", should be re-design and/or changed colors and make it more attractive and informative.

Reply: Many thanks for your advice. We will re-design this figure in the revision to emphasize the result of the most optimal policy. Also, the figure color will be changed to be more attractive as follows.

[Figure]

Fig. 10 Annual hydropower increment of seven pools (A-G) for the most optimal policy with *COR* (Unit: billion kWh).

(6) Some jargon in the reservoir community is not accessible to all audiences of HESS, ex., 'Inflow stochasticity' in implicit stochastic optimization (ISO), explicit stochastic optimization (ESO), and parameter simulation optimization (PSO) in the Introduction.

Reply: Many thanks for your review, considering also the similar comments with Reviewer 1, we fully agree that the explanation of jargon needs a careful improvement to serve the wide community of audiences of HESS. We will carefully rewrite this part in the revision following the advices of both reviewers.

(7) The references could be updated.

Reply: Thanks for your comments. We will update the references with some recent and related literatures.

(8) Proofreading by a native English speaker should be conducted to improve both language and organization quality.

Reply: Many thanks for your comments. We will carefully check the revised version, and we will also look for professional language editing service for proofreading and language improvement before submitting the revised version.